# Cretaceous bird from Brazil informs the evolution of the avian skull and brain

Luis M. Chiappe[1,8 ✉], Guillermo Navalón[2,8 ✉], Agustín G. Martinelli[3], Ismar de Souza Carvalho[4,5], Rodrigo Miloni Santucci[6], Yun-Hsin Wu[1] & Daniel J. Field[2,7 ✉]

A dearth of Mesozoic-aged, three-dimensional fossils hinders understanding of the origin of the distinctive skull and brain of modern (crown) birds[1]. Here we report *Navaornis hestiae* gen. et sp. nov., an exquisitely preserved fossil species from the Late Cretaceous of Brazil. The skull of *Navaornis* is toothless and large-eyed, with a vaulted cranium closely resembling the condition in crown birds; however, phylogenetic analyses recover *Navaornis* in Enantiornithes, a highly diverse clade of Mesozoic stem birds. Despite an overall geometry quantitatively indistinguishable from crown birds, the skull of *Navaornis* retains numerous plesiomorphies including a maxilla-dominated rostrum, an akinetic palate, a diapsid temporal configuration, a small cerebellum and a weakly expanded telencephalon. These archaic neurocranial traits are combined with a crown bird-like degree of brain flexion and a bony labyrinth comparable in shape to those of many crown birds but substantially larger. Altogether, the emergent cranial geometry of *Navaornis* shows an unprecedented degree of similarity between crown birds and enantiornithines, groups last sharing a common ancestor more than 130 million years ago[2]. *Navaornis* provides long-sought insight into the detailed cranial and endocranial morphology of stem birds phylogenetically crownward of *Archaeopteryx*, clarifying the pattern and timing by which the distinctive neuroanatomy of living birds was assembled.

Enantiornithes is the most species-rich and ubiquitously distributed clade of Mesozoic birds, known from every continent except Antarctica[2]. Nonetheless, as a result of taphonomic flattening associated with most enantiornithine-bearing localities (for example, the Jehol, Las Hoyas and Araripe Lagerstätten), complete, three-dimensionally preserved skulls are rare[3–7]. Virtually no well-preserved skulls bearing undistorted endocasts are known from taxa phylogenetically intermediate between *Archaeopteryx* (the earliest known Mesozoic bird) and some of the closest known relatives of crown birds (for example, *Ichthyornis* and *Hesperornis*), a gap exceeding 60 million years and encompassing most of the phylogenetic history of Mesozoic birds[7–11]. As a result, questions about the origins of the derived skulls and brains of crown birds remain among the most tantalizing knowledge gaps in vertebrate macroevolution[1,7] and the detailed cranial osteology and endocranial morphology of enantiornithines remain poorly known[3,12].

A bonebed contained in the Sítio Paleontológico 'José Martin Suárez' in Presidente Prudente (São Paulo State, southeastern Brazil) represents the richest Late Cretaceous fossil bird locality known and has exceptional potential to provide insight into the cranial morphology of enantiornithines. Unlike other quarries yielding an abundance of enantiornithine remains, such as those from the Early Cretaceous Jehol biota, this quarry is small in size (most fossils are contained in an area of about 6 m² and in a sedimentary layer less than 50 cm deep), providing an unprecedented snapshot of a Late Cretaceous terrestrial avifauna. The quarry has thus far yielded hundreds of three-dimensionally preserved enantiornithine bones. Here we report on the only complete avian skull from this locality, which constitutes the best-preserved Mesozoic bird skull yet discovered, yielding insights into how and when characteristic features of the modern bird skull and central nervous system evolved. The remarkable state of preservation of the fossil may also clarify anatomical interpretations of previously known flattened enantiornithine cranial remains[3].

## Systematic palaeontology

Aves Linnaeus, 1758
Ornithothoraces Chiappe and Calvo, 1994
Enantiornithes Walker, 1981
*Navaornis hestiae* gen. et sp. nov.

**Remarks.** We use Aves to refer to all taxa descended from the most recent common ancestor of *Archaeopteryx lithographica* and crown birds[13].
**Etymology.** *Navaornis* honours William Nava, who discovered the fossil locality in 2004 and the holotype specimen in 2016; the specific epithet *hestiae* alludes to Hestia, the Greek goddess of architecture, regarded as simultaneously the oldest and the youngest of the Twelve

[1]Dinosaur Institute, Natural History Museum of Los Angeles County, Los Angeles, CA, USA. [2]Department of Earth Sciences, University of Cambridge, Cambridge, UK. [3]Sección Paleontología de Vertebrados, CONICET—Museo Argentino de Ciencias Naturales Bernardino Rivadavia, Buenos Aires, Argentina. [4]Instituto de Geociências, Universidade Federal do Rio de Janeiro, Rio de Janeiro, Brazil. [5]Centro de Geociências, Coimbra University, Coimbra, Portugal. [6]Faculdade UnB Planaltina, Universidade de Brasília, Brasília, Brazil. [7]Museum of Zoology, University of Cambridge, Cambridge, UK. [8]These authors contributed equally: Luis M. Chiappe, Guillermo Navalón. ✉e-mail: lchiappe@nhm.org; gn315@cam.ac.uk; djf70@cam.ac.uk

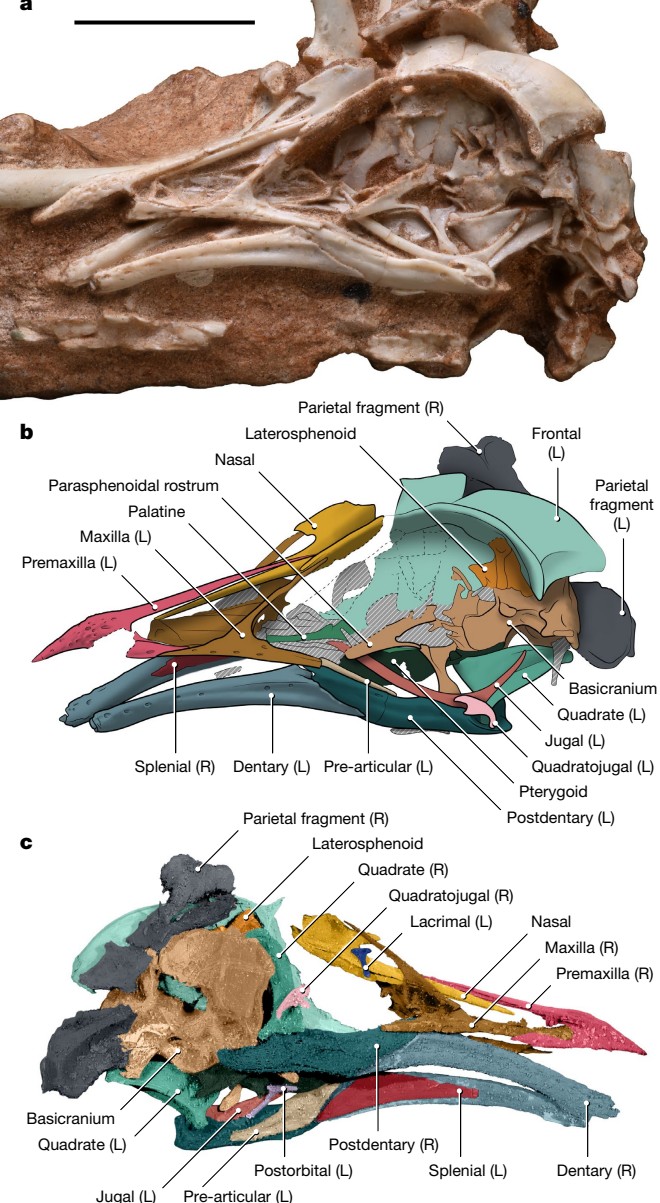

**Fig. 1 | The holotype of the enantiornithine *N. hestiae* gen. et sp. nov. from the Late Cretaceous of Brazil. a,b**, Photograph (**a**) and interpretive drawing (**b**) of the exposed side of the holotype of *N. hestiae* (MPM-200-1) in left lateral view. **c**, Micro-computed tomography rendering of MPM-200-1 in right ventral–lateral view. Scale bar, 10 mm.

Olympians. *Navaornis* reflects this duality in that it belongs to an archaic lineage, yet its cranial geometry is essentially modern.

**Holotype.** MPM-200-1 (MPM, Museu de Paleontologia de Marília, Marília, Brazil). A complete skull (Fig. 1) articulated with the anterior-most cervical vertebrae, extracted from a block (MPM-200; Extended Data Fig. 1) from the William's Quarry bonebed at Sítio Paleontológico 'José Martin Suárez'. A cast of MPM-200 has been accessioned at the Dinosaur Institute, Natural History Museum of Los Angeles County.

**Referred specimens.** MPM-334-1, an isolated basicranium from William's Quarry at Sítio Paleontológico 'José Martin Suárez'[12] whose morphology is identical to that preserved in MPM-200-1. MPM-200 includes a partially articulated postcranial skeleton, which is also referred to *N. hestiae* and is thought to belong to the same individual as the skull (Extended Data Fig. 1).

**Locality and age.** William's Quarry is contained within the Adamantina Formation (Bauru Group, Bauru Basin); various lines of evidence[14–17] indicate a late Santonian to early Campanian age (around 85–75 million years ago) for this site.

**Diagnosis.** Enantiornithine with a toothless skull and a combination of the following features: fully fused premaxillae with a convex dorsorostral surface, highly curved jugal, small, comma-shaped quadratojugal, diminutive lacrimal failing to separate the orbit from the antorbital fenestra, parasphenoidal rostrum perforated by a large ovoid fenestra, elongate basipterygoid processes, large and sinusoidal anterior semicircular canal excavating the dorsal margin of the supraoccipital, robust and prominent medial process of the mandible.

**Phylogeny.** Phylogenetic analyses (Methods) recovered the holotype of *N. hestiae* (MPM-200-1) within Enantiornithes, regardless of whether characters from the referred postcranium were included (Extended Data Fig. 2; see 'Data availability' for all phylogenetic tree topologies). Specifically, *Navaornis* clusters with *Gobipteryx* and *Yuornis*, other edentulous Late Cretaceous enantiornithines, although we note that most internal relationships within Enantiornithes remain poorly resolved[18] and this last result may be influenced by a propensity for homoplastic tooth loss in Mesozoic birds[19]. The minimum age for the divergence between the most recent common ancestor of Enantiornithes and their sister clade Euornithes, which includes crown birds[20,21] (the Ornithothoraces node), is defined by the oldest known, bona fide enantiornithines and euornithines from the lacustrine deposits of the Huajiying Formation (Jehol biota, northeastern China), dated to about 131 million years ago[22,23].

## Morphological description

The jaws of MPM-200-1 are completely toothless (Fig. 1). The premaxillae are fused into a single element, an unusual condition among enantiornithines, shared with *Shangyang*[24] and other enantiornithines from William's Quarry[25], and the frontal process tapers caudally, terminating between the nasals without reaching the frontals, in contrast to the condition in *Yuornis*[26] (Figs. 1 and 2 and Extended Data Figs. 2–6). The maxillary process of the premaxilla tapers caudally, inserting into a shallow trough on the lateral side of the maxilla (Figs. 1 and 2). The inverted T-shaped maxilla constitutes a prominent component of the facial margin (at least 1.5 times the length of the premaxilla at the tomial margin; Figs. 1 and 2). The maxillary rami are far more gracile than in many Early Cretaceous enantiornithines (for example, Bohaiornithidae[27], *Shenqiornis*[28] and *Pengornis*[29]), even when skeletally immature specimens are considered[30,31]. The premaxillary ramus of the maxilla is subequal in length to the jugal ramus and both are more robust than the nasal ramus; the last is long and directed caudomedially, underlying the maxillary process of the nasal and forming a medially directed lamina along its basal two-thirds (Figs. 1 and 2 and Extended Data Fig. 6). Medially, the premaxillary ramus expands into a sheet of bone which forms the palatal roof, as in *Gobipteryx*[32] (Fig. 2 and Extended Data Figs. 2, 5 and 6). The confluence of the nasal and jugal rami form the rostral margin of the antorbital fenestra, which exhibits a funnel-like recess probably of pneumatic origin[33] (Extended Data Fig. 6). The naris is elongate and tear-shaped, tapering caudodorsally; its caudal margin is formed by the maxilla and nasal, as in other Enantiornithes[3] (Figs. 1 and 2 and Extended Data Figs. 2 and 5).

The rostral two-thirds of the nasal comprises an elongate prong, which laterally lines the fused frontal process of the premaxilla; the postnarial portion of the nasal is proportionately shorter than in other enantiornithines (Figs. 1 and 2 and Extended Data Figs. 2–6). A minute, T-shaped bone is regarded as the lacrimal, an interpretation supported by its position between the orbit and the antorbital fossa and its overall morphology. Its ventral ramus curves slightly medially and, as in *Yuornis*, does not separate the orbit from the broad antorbital fenestra (Figs. 1c and 2 and Extended Data Figs. 2 and 5). The jugal is elongate and

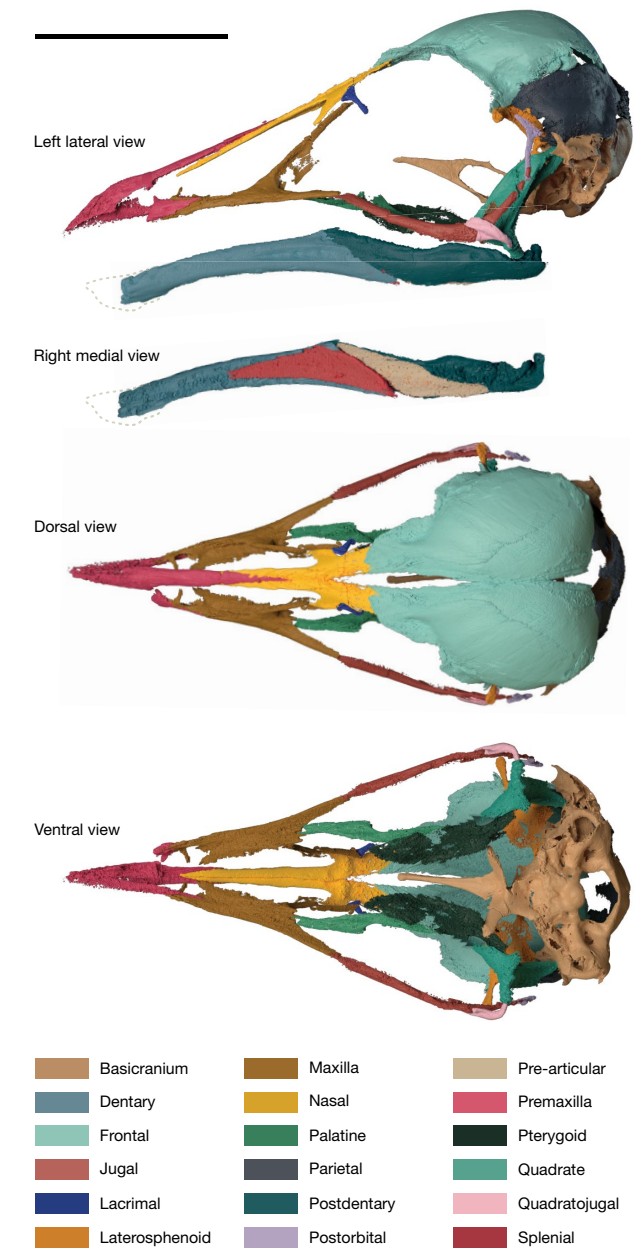

Left lateral view

Right medial view

Dorsal view

Ventral view

| | | |
|---|---|---|
| Basicranium | Maxilla | Pre-articular |
| Dentary | Nasal | Premaxilla |
| Frontal | Palatine | Pterygoid |
| Jugal | Parietal | Quadrate |
| Lacrimal | Postdentary | Quadratojugal |
| Laterosphenoid | Postorbital | Splenial |

**Fig. 2 | Three-dimensional cranial reconstruction of *N. hestiae*.** Composite three-dimensional reconstruction of the skull of *N. hestiae* from MPM-200-1 and referred braincase MPM-334-1. Scale bar, 10 mm.

strongly bowed, with a long, tapering postorbital ramus and a greatly reduced quadratojugal ramus (Figs. 1 and 2 and Extended Data Figs. 2, 3, 5 and 6). This morphology is reminiscent of that seen in some other Enantiornithes[30,31,34], although the postorbital process of *Navaornis* is proportionately longer and the quadratojugal ramus proportionately shorter. Another T-shaped bone, twice the size of the one interpreted as the lacrimal, is preserved in contact with the medial surface of the left jugal and is interpreted as a postorbital (Fig. 1c and Extended Data Figs. 2 and 4), an element previously identified in some enantiornithine taxa[3,31,34]. It exhibits a short squamosal ramus and a longer frontal ramus (Extended Data Fig. 6), presumably connecting to the postorbital process of the frontal (Fig. 2b). The length of the rami of the postorbital and jugal suggest that these bones would have fully separated the orbit from an infratemporal fenestra. A short, comma-shaped bone is interpreted as the quadratojugal (Figs. 1 and 2 and Extended Data Figs. 2–6); it differs from the flattened L-shaped quadratojugal known from some

other enantiornithines[31,35,36]. Although disarticulated, the left and right quadratojugals are preserved in similar positions on both sides of the skull, implying that they are close to their natural positions (Fig. 1b,c), probably contacting the quadrate caudoventrally (Figs. 1 and 2 and Extended Data Fig. 5). Additionally, the quadratojugal exhibits a short anteroventral ramus which projects from the midpoint of the bone, a feature not observed in other enantiornithines[31,35,36].

The frontal is large and vaulted. It exhibits a straight interfrontal margin, clearly unfused to its counterpart, similar to the condition in *Yuornis*[26], *Zhouornis*[37] and other enantiornithines[34], as well as isolated frontals recovered from the same outcrop; its caudal margin is similarly unfused to the parietal, which bears a distinct nuchal crest (Extended Data Figs. 2, 6 and 7). Caudolaterally, the frontal projects into a triangular postorbital process, also present in *Zhouornis*[37] and the Montsec immature enantiornithine[30]. The postorbital process of the frontal seems to overlie a slightly larger, laterally projecting postorbital process of the laterosphenoid, as interpreted in *Ichthyornis*[9] (Fig. 2). The internal surface of the laterosphenoid bears an ample fossa circumscribing part of the optic lobe, with the remainder of the depression extending caudally onto the parietal (Extended Data Fig. 4). The basicranium (including the vestibular region) is virtually identical to a previously reported braincase from the same quarry[12] (for example, bearing large and slender basipterygoid processes and a long parasphenoidal rostrum perforated by an ovoid fenestra), which we here refer to *N. hestiae* (Extended Data Fig. 8). This referred specimen (MPM-334-1) bears a triradiate squamosal with a tapering zygomatic process similar to the Montsec immature enantiornithine[30] which seems to be partially fused to the rest of the basicranium (Extended Data Fig. 8).

Information from the postorbital, jugal, quadratojugal, squamosal and laterosphenoid allows the skull of *Navaornis* to be reconstructed with a diapsid temporal configuration; the infratemporal fenestra is unbounded caudally as the squamosal does not seem to contact the quadratojugal (Fig. 2 and Extended Data Figs. 2 and 5). Varying degrees of closure of the upper and infratemporal fenestrae have been previously reported for Enantiornithes[30,31,36,38] as well as both crownward[9] and stemward[39,40] Mesozoic birds. In *Navaornis*, although the postorbital probably contacted the caudal portion of the jugal, a continuous bony connection between the postorbital and the caudally positioned zygomatic process of the squamosal (incomplete or weathered in both the holotype and referred specimens) would have been very delicate or partially ligamentous (Figs. 1b and 2 and Extended Data Fig. 8).

Portions of the palate are infilled with radiopaque minerals, thus only the morphology of the left palatine and pterygoid can be unambiguously discerned from our computed tomography scans (Fig. 2 and Extended Data Figs. 3–6). The palatine is rostrocaudally elongate and mediolaterally narrow, with a rostrally directed maxillary ramus, a dorsally projecting flange at its rostrocaudal midpoint and a dorsomedially slanted pterygoid ramus; these features are reminiscent of those of *Gobipteryx*[32], although the palatine of *Navaornis* is proportionately longer and more gracile. The pterygoid has a broad, lateroventrally slanted body and is roughly oval in shape. It has an expanded and biradiate caudal margin, including a prominent dorsolateral ramus which develops an ample contact with the orbital process of the quadrate (Extended Data Fig. 6). The morphology of the quadrate ramus of the pterygoid is comparable to that of a previously described juvenile enantiornithine[31] and many non-avian theropods (for example, refs. 41,42) and the pterygoid lacks a mobile connection with the palatine/hemipterygoid as has been reported in the Cretaceous ornithurines *Ichthyornis* and *Janavis*[11,43].

The quadrate exhibits a broad orbital process (Figs. 1 and 2 and Extended Data Figs. 5 and 6), as is typical of non-ornithothoracine avians, Enantiornithes and stemward Euornithes (for example, *Patagopteryx*[44]). Its ventrolateral corner exhibits a small, circular quadratojugal cotyle (Fig. 2 and Extended Data Figs. 5 and 6); although similar cotyles of varying sizes are known in Cretaceous Euornithes (for example,

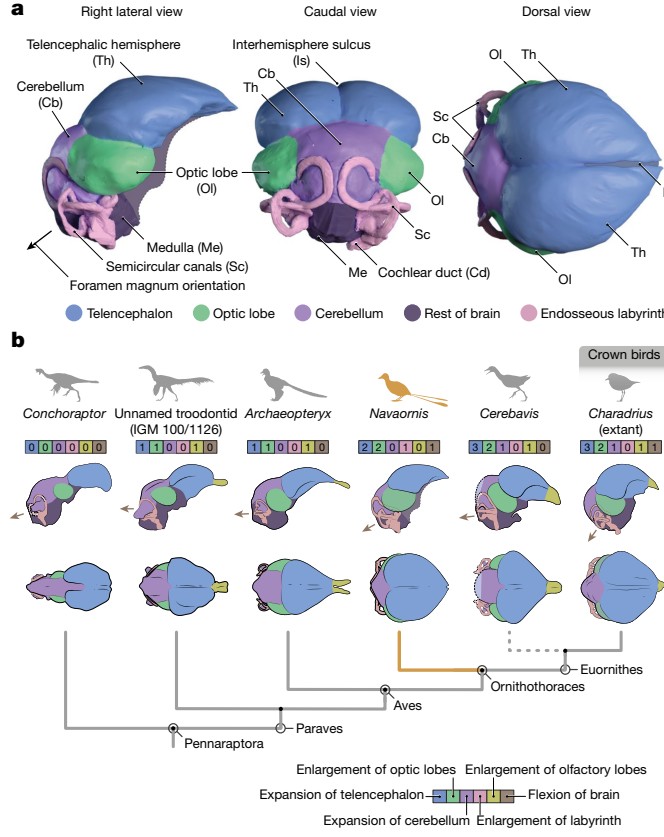

**a**

Right lateral view

Telencephalic hemisphere (Th)

Cerebellum (Cb)

Optic lobe (Ol)

Medulla (Me)

Semicircular canals (Sc)

Foramen magnum orientation

Caudal view

Interhemisphere sulcus (Is)

Cb

Th

Ol

Sc

Me    Cochlear duct (Cd)

Dorsal view

Ol    Th

Sc

Cb

Ol

Is

Th

● Telencephalon   ● Optic lobe   ● Cerebellum   ● Rest of brain   ● Endosseous labyrinth

**b**

Crown birds

*Conchoraptor*   Unnamed troodontid (IGM 100/1126)   *Archaeopteryx*   *Navaornis*   *Cerebavis*   *Charadrius* (extant)

`0 0 0 0 0 0`   `1 1 0 0 0 1`   `1 1 0 0 1 0`   `2 2 0 1 0 1`   `3 2 1 0 1 0`   `3 2 1 0 1 0`

Euornithes

Ornithothoraces

Aves

Paraves

Pennaraptora

Enlargement of optic lobes   Enlargement of olfactory lobes

Expansion of telencephalon ⬛🟨🟪🟩🟥⬛ Flexion of brain

Expansion of cerebellum   Enlargement of labyrinth

**Fig. 3 | Endocranial anatomy of *N. hestiae* and brain evolution in stem birds.** **a**, Three-dimensional reconstruction of the endocranial morphology of *N. hestiae* from MPM-200-1 and MPM-334-1. Portions deriving from MPM-200-1 and MPM-334-1, as well as the reconstruction process, are explained in the Methods and Extended Data Fig. 7. **b**, Evolution of endocranial morphology across Pennaraptora. Numbers in the coloured boxes refer to the degree of expansion of each of the main neuroanatomical and sensorial regions for each taxon. Brown arrows in **b** depict the orientation of the foramen magnum.

*Patagopteryx*[44], *Parahesperornis*[45], *Hesperornis*[46] and *Ichthyornis*[9,47]), this trait has never previously been reported in Enantiornithes.

The dentary occupies more than half the rostrocaudal length of the mandible (Fig. 1). It bears an obliquely oriented suture with the postdentary complex, as in other Enantiornithes. The robust medial process of the lower jaw projects dorsomedially, as in *Yuornis*[26] and possibly *Gretcheniao*[18]. Most sutures in the postdentary complex are not distinguishable, although a triangular splenial and a broad pre-articular line the lingual margin of the caudal half of the mandible (Figs. 1 and 2), similar to the condition in some extant anseriforms.

Nearly complete endocranial reconstructions of the brain and inner ear of *Navaornis* were generated from the well-preserved frontals, parietals, laterosphenoids and basicranium of MPM-200-1 and MPM-334-1 (Methods; Fig. 3a and Extended Data Figs. 7 and 8). The brain is heart-shaped in dorsal view and exhibits a flexed configuration with the brainstem projecting ventrally to a degree heretofore seen only in crown birds[12,48] (Fig. 3). The extent of separation between the telencephalic hemispheres remains somewhat ambiguous given that only the left frontal is well preserved. The hemispheres are vaulted and pyriform in dorsal view and are mediolaterally expanded to a degree exceeding that of *Archaeopteryx*[7,49] and known non-avian pennaraptoran dinosaurs[50,51] (Fig. 3b). However, the hemispheres do not envelop the dorsal surface of the optic lobes, as in *Cerebavis* and most crown birds, in which the lobes are obscured in dorsal view by extreme mediolateral expansion of the telencephalon[52]. Unlike the condition in *Ichthyornis*[11] and most crown birds, the dorsal surface of the telencephalon is smooth

(Fig. 3b), lacking a discernible Wulst—a thickening of the hyperpallium often visible on the telencephalic surface and an important structure involved in integrating visual and somatosensory information[52]. The optic lobe is subspherical and lies immediately ventral to the posterior portion of the telencephalon; both structures terminate caudally at the same point and probably have an extended contact area as a result of the ventral expansion of the telencephalon (Fig. 3). The optic lobe is more caudally positioned than in any previously known stem bird[12], approaching the degree seen in many crown birds with moderately to strongly flexed brains[52]. The optic lobe sits dorsal to a greatly enlarged and sinusoidal anterior semicircular canal of the endosseous labyrinth whose morphology is indistinguishable from that of the referred specimen MPM-334-1 (ref. 12) (Extended Data Fig. 8). The cerebellum seems to be small, lacking the inflated morphology of *Cerebavis* and many crown birds, indicative of its volumetric expansion[7,8] (Fig. 3). Indeed, the cerebellum of *Navaornis* seems to be even more weakly developed than those of *Archaeopteryx*[7,49] and some non-avian dinosaurs (for example, *Zanabazar*[53] and IGM 100/1126 (ref. 1)) and no distinct foliation of the cerebellar surface is discernible[54,55]. The endocast of the medulla is distorted in MPM-200-1 but its globular morphology matches that of the referred specimen MPM-334-1 (ref. 12).

## Discussion

Combined with recently published material[25], *Navaornis* provides evidence of co-occurring toothless and toothed enantiornithines from a single site, hinting at a considerable degree of sympatric ecological disparity in some Cretaceous bird communities. The toothless condition of its skull notwithstanding, the plesiomorphic nature of most individual elements in the skull of *Navaornis* (Fig. 2 and Extended Data Figs. 5 and 6) supports the suggestion that enantiornithines retained an akinetic skull with a diapsid temporal configuration[30,36]. Nonetheless, the geometry of the three-dimensionally preserved and undistorted skull of *Navaornis* is similar to that of crown birds overall. This interpretation is supported by our quantitative morphometric comparisons which show that *Navaornis* falls comfortably in a densely occupied region of extant avian cranial morphospace defined by the three principal components of cranial shape variation (Fig. 4a and Extended Data Fig. 10). *Navaornis* is closest in overall cranial geometry to an array of extant taxa belonging to disparate crown bird groups (for example, *Chauna*, *Psophia*, *Chunga* and *Corvus*; Extended Data Fig. 9).

*Navaornis* provides a long-awaited illustration of the morphology of the brain and inner ear within a critically undersampled phylogenetic interval of the avian stem lineage, capturing a previously unknown combination of plesiomorphic and derived traits which clarify the pattern by which the distinctive central nervous system of living birds arose (Fig. 3b). Our quantitative geometric comparisons demonstrate that *Navaornis* exhibits a brain morphology intermediate between *Archaeopteryx* and crown birds along the main axis of endocranial shape variation (PC1), which accounts for nearly half of total neuroanatomical shape variation in birds and closely related theropod dinosaurs (Fig. 4b and Extended Data Fig. 10). Specifically, *Navaornis* suggests that the expansions of the telencephalon and, to a lesser extent, cerebellum characteristic of crown birds[7,56] seem to have arisen at a comparatively crownward point in avian evolutionary history (Figs. 3b and 4b). The characteristic inflated morphology of the cerebellum in crown birds can be concealed by ventral folding (flexion) of the brain in small-sized neoavian birds (for example, hummingbirds, woodpeckers and passerines)[52], potentially leading to underestimates of cerebellar size from endocranial surface reconstructions. However, although the endocast of *Navaornis* is more ventrally flexed than that of many extant representatives of Palaeognathae and Galloanserae[57], it is substantially less flexed than in most extant neoavian birds[12], suggesting that our reconstruction of a comparatively flat cerebellum in *Navaornis* is unlikely to be meaningfully influenced by brain flexion.

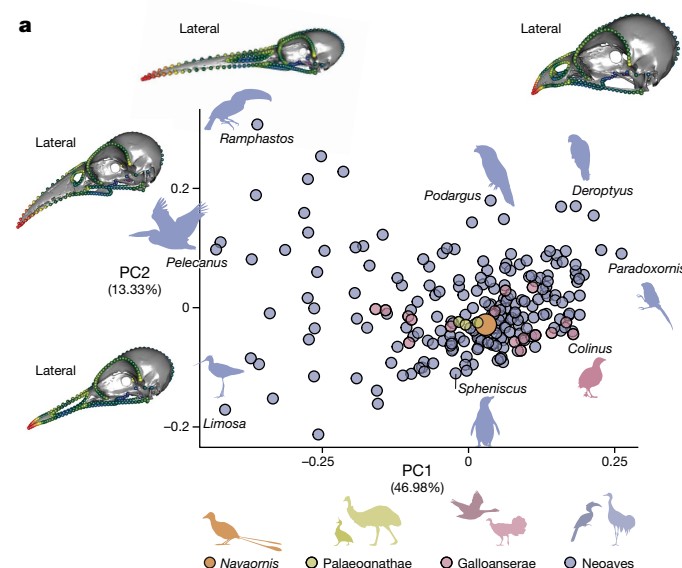

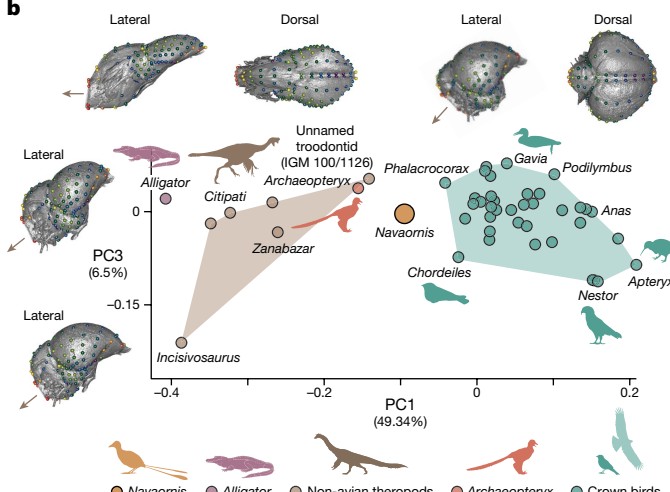

**Fig. 4 | *Navaornis* combines a geometrically crown bird-like skull with a central nervous system morphologically intermediate between *Archaeopteryx* and crown birds. a**, Three-dimensional principal component morphospace of neornithine skulls (PC1 versus PC2). Warped models illustrate extremes along PC1 and PC2. **b**, Three-dimensional principal component morphospace (PC1 versus PC3) of endocrania from Aves and relevant non-avian taxa. *Navaornis* falls in crown bird morphospace along PC3 but not along PC2 (Extended Data Fig. 10). Landmarks are colour-coded for high (warm colours) to low (cold colours) per-landmark variances. Brown arrows in **b** depict the foramen magnum orientation.

The presence of a ventrally flexed brain configuration, caudally displaced optic lobes and an enlarged sinusoidal labyrinth in *Navaornis* implies that the origins of these 'advanced' traits often associated with crown birds[11,58] either predated the origin of Ornithothoraces or evolved convergently among both Enantiornithes and crownward Euornithes. Regardless of which of these evolutionary scenarios is supported by future findings, aspects of the neuroanatomy of *Navaornis* are conspicuously similar to those of late-stage extant bird embryos and hatchlings (for example, *Gallus* and *Ficedula*[59]). For example, the optic tectum of crown birds attains adult-like proportions in early ontogeny along with a ventrally oriented brainstem in species with a ventrally flexed brain, yet the telencephalon and cerebellum only acquire their more expanded forms later in development, echoing other instances of ontogenetic and phylogenetic covariation across

the dinosaur–bird transition[53,60,61]. Altogether, the morphology of the endocast of *Navaornis* shows an intermediate stage in the evolutionary history of the unique avian brain and its combination of crown bird-like and plesiomorphic attributes is congruent with hypotheses of modular evolution whereby different anatomical regions evolve quasi-independently at varying rates[60].

In *Navaornis*, this intermediate stage of brain evolution is combined with a skull retaining numerous plesiomorphic features in the main cranial elements, which serve as building blocks from which a geometrically modern cranial configuration is constructed. This degree of geometric convergence between Enantiornithes and crown birds suggests that developmental constraints responsible for canalizing the general shape of the bird skull may have been present throughout much of avian evolutionary history, predating both the phylogenetic divergence between Enantiornithes and Euornithes more than 130 million years ago as well as the evolutionary acquisition of several apomorphic characteristics of crown bird skull and brain morphology. The exceptionally well-preserved skull of *Navaornis* emphasizes the necessity of hitherto elusive undistorted Mesozoic bird skulls for illuminating the complex sequence by which the unique brains and skulls of modern birds arose.

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

## Methods

### Preparation and imaging of specimens

MPM-200-1 was scanned using a Bruker SkyScan 1173 computed tomography scanner at the Instituto de Petróleo e dos Recursos Naturais (Laboratório de Sedimentologia e Petrologia) of the Pontifícia Universidade Católica do Rio Grande do Sul (PUCRS), Porto Alegre, Rio Grande do Sul, Brazil. Scanning parameters were as follows: 10.71 μm voxel size, 60 kV, 133 μA, exposure time 850 ms, averaging two frames, no 360° rotation, using an aluminium filter of 1.0 mm. Slices were reconstructed using the software NRecon v.1.7.4.6. Volumes were digitally reconstructed and segmented using VGSTUDIOMAX 3.4.0 (VolumeGraphics; see Extended Data Figs. 3 and 4 for all segmented elements in situ).

### Three-dimensional skull and endocast reconstruction

Three-dimensional meshes of each recognizable segmented cranial element and endocranial surface were exported from VGSTUDIOMAX 3.4.0 to Blender 3.3.0, in which they were re-articulated to reconstruct the skull and endocranial anatomy of *N. hestiae* (Figs. 2 and 3a and Extended Data Figs. 5 and 7). Owing to the lack of distortion of most cranial elements, only the left frontal bone required slight retrodeformation following established best practices[62]. This was accomplished using the Lattice function in Blender, in which mediolateral compression of this element was corrected to match the geometry of the dorsal rim of the complete and mostly undistorted left parietal bone (Extended Data Fig. 7). The same degree of retrodeformation was then applied to the endocast surface of the left frontal, that is, the left hemisphere of the telencephalon (Extended Data Fig. 7). The remaining surfaces composing the endocranium of *N. hestiae* were: ventral (derived from the basicrania of both the holotype and referred specimen MPM-334-1), anterior (derived from the right and left laterosphenoids of the holotype), dorsal and lateral surfaces of the left optic lobes (derived from the parietal of the holotype), cerebellum (derived from the parietal of the holotype and basicranium of both the holotype and referred specimen MPM-334-1) and medulla (derived from both the holotype and referred specimen MPM-334-1; Extended Data Fig. 7). Left and right surfaces were mirrored from the best-preserved element/endocranial surface.

### Three-dimensional geometric morphometrics of the skull and endocast

Landmark-based geometric morphometrics were used to quantitatively compare the exocranial and endocranial anatomy of *Navaornis* with crown birds and a selection of non-avian dinosaurs and Mesozoic birds. First, the three-dimensional geometry of our exocranial reconstruction of *Navaornis* (Fig. 2 and Extended Data Figs. 5 and 7) was captured by adapting a previously proposed landmarking scheme designed for crown birds[63] to enantiornithine cranial morphology (Extended Data Fig. 10), enabling us to explore the striking similarities in exocranial geometry between crown birds and *Navaornis* despite their anatomical differences across several regions of the skull. For instance, although the maxilla represents one of the main elements of the rostrum in enantiornithines, it is greatly reduced in crown birds, drawing the cranial end of the jugal rostrally such that it is essentially level with the rostral-most point of the antorbital fenestra. To accommodate these anatomical differences between crown birds and enantiornithines, we moved the landmark located at the rostral end of the jugal bar in crown birds from the original landmarking scheme to the rostral-most point of the antorbital fenestra on the ventrolateral rim of the maxilla in *Navaornis* (Extended Data Fig. 10). Likewise, we redefined the original landmark at the caudoventral-most point of the maxilla and placed it just rostral to the aforementioned modified landmark in *Navaornis*. These modifications affected the semilandmark curves defining the ventrolateral rims of the upper jaw and jugal bar, as these strings of semilandmarks are anchored to the aforementioned landmarks. These semilandmarks were redigitized ensuring that they

were evenly spaced along the redefined curves. Finally, the landmark defining the basioccipital–parabasisphenoidal rostrum contact in the original landmarking scheme was placed just caudal to the recess representing the caudal-most point of the parasphenoidal rostrum in *Navaornis*. Following these modifications to the landmarking scheme, we incorporated the three-dimensional coordinates of *Navaornis* into a broad three-dimensional avian exocranial morphological dataset encompassing 228 extant species representing the phylogenetic breadth of crown birds[64].

We then applied a previously published landmarking scheme[56] to the endocranial reconstruction of *Navaornis* (Extended Data Fig. 10) to capture and compare its three-dimensional endocranial anatomy with a representative sample of crown birds, *Archaeopteryx*, closely related non-avian theropods and *Alligator*, representing Crocodylia, the extant sister group to birds. Exocranial landmarks were digitized in Avizo Lite 2019.3 (Thermo Fisher Scientific) and endocranial landmarks were digitized in Landmark Editor[65] following previously described procedures[56,64]. Thereafter, the landmark datasets were imported into the R statistical environment v.4.1.2 (ref. 66), in which all downstream analyses were conducted.

Generalized Procrustes analyses were performed on both sets of landmark coordinates to separate shape data from size and other confounding factors and the minimum bending energy criterion[67] was used to slide curve (exocranium and endocranium) and patch (endocranium) semilandmarks following previously described procedures[56,64], using the function gpagen in the R package geomorph v.4.0.5 (ref. 68).

Principal components analyses were carried out on the exocranial and endocranial Procrustes coordinates to visualize shape variation using the function gm.prcomp in geomorph. To determine the extant species geometrically closest to *Navaornis* in exocranial shape, we determined the Procrustes distances between *Navaornis* and all extant taxa in our dataset (Extended Data Fig. 9) using Euclidean distances with the function dist from the R package stats v.4.1.2 (ref. 66). Changes associated with main axes of exocranial and endocranial shape variation were illustrated as deformations warped from the three-dimensional surface of the exocranium and endocranium of the individual species closest to the mean shape in both samples. Specifically, this three-dimensional surface and the mean shape from the sample were projected onto the scores representing the 0.05 and 0.95 quantiles for each principal component axis by means of thin-plate spline deformation[69] using the function tps3d from the package Morpho v.2.10 (ref. 70) and shape. predictor from geomorph. We also plotted the respective landmark configurations onto the deformed meshes using shape.predictor and coloured these landmark constellations according to per-landmark variances from each dataset using the hot.dots function (freely available following this link: https://zenodo.org/record/3929193)[71].

### Phylogenetic analysis

Heuristic parsimony analyses were applied to a previously published and expanded (adding *Navaornis* and *Yuornis*) dataset[18] using TNT v.1.6 (ref. 72) under equal and implied weights ($K$ = 3, 9 and 12). Twenty-four multistate characters (1, 3, 8, 23, 39, 44, 48, 51, 52, 68, 77, 84, 134, 143, 149, 162, 173, 180, 181, 183, 187, 190, 195 and 200) were treated as additive (or 'ordered'), according to the recommendation that multistate characters be analysed as additive when they represent morphoclines (for example, small–medium–large)[73]. We analysed a character matrix in which the scorings for *Navaornis* were based exclusively on the holotype (MPM-200-1) and another matrix with further scorings from the referred specimen MPM-200 (a postcranial skeleton contained in the same block; Extended Data Fig. 1). Analyses were performed with 10,000 replicates, using tree bisection–reconnection branch swapping, retaining ten trees per replicate. Support was obtained by calculating bootstrap values set at 10,000 replicates. The equal weights analysis using only cranial scorings resulted in 102 most parsimonious trees of 909 steps each; the strict consensus tree resulted in a large

polytomy among enantiornithines. By contrast, the analysis using $K = 3$ (fit = 88.25) resulted in two most parsimonious trees, whereas those performed under $K = 9$ (fit = 46.21) and $K = 12$ (fit = 37.53) resulted in one tree each with nearly identical topologies. The consensus tree obtained from the $K = 3$ analysis has some polytomies in Enantiornithes, whereas the relationships in this clade in the $K = 9$ and 12 trees are fully resolved. The analysis of the character matrix with further postcranial scorings resulted in two most parsimonious trees of 916 steps and yielded a nearly identical topology to the analysis including only the holotype skull. In all most parsimonious trees, *Navaornis* is nested in the enantiornithine clade. Trees resulting from all the analyses are freely available at ref. 74.

### Reporting summary

Further information on research design is available in the Nature Portfolio Reporting Summary linked to this article.

## Data availability

Scan data and surface meshes of all preserved elements of *Navaornis* are housed on MorphoSource (https://www.morphosource.org/projects/000608371?locale=en). Phylogenetic matrices and morphometric landmark coordinates are available at Zenodo (https://doi.org/10.5281/zenodo.10696014)[74]. The Life Science Identifier for *N. hestiae* is urn:lsid:zoobank.org:act:BF806CE5-CD23-45AB-BCB0-93193D4FE378.

## Code availability

The code underpinning our morphometric analyses is available at Zenodo (https://doi.org/10.5281/zenodo.10696014)[74].

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

**Acknowledgements** This study is the product of a multiyear, international collaboration between the Museu de Paleontologia de Marília (MPM), the Museo Argentino de Ciencias Naturales and the Natural History Museum of Los Angeles County (NHMLAC). We are extremely grateful to W. Nava (MPM), who collected the specimens described herein, as well as many other extraordinary fossils from the Sítio Paleontológico 'José Martin Suárez'. The Prefeitura Municipal de Marília, the Secretaria de Cultura and Secretaria de Trabalho, Turismo e Desenvolvimento Econômico, the Prefeitura Municipal de Presidente Prudente (former mayor N. Bugalho, former technical assistant I. Cavalcante, current mayor E. Thomas and former municipal secretary of tourism A. Padilha) and the Agência Nacional de Mineração for the authorization to collect the fossils (no. 026/2015, process no. 001.058/2015); all provided essential support during fieldwork and laboratory activities. We are also grateful to the Instituto de Petróleo e dos Recursos Naturais (Laboratório de Sedimentologia e Petrologia) of PUCRS for undertaking microtomography scans of the material. We thank S. Abramowicz (NHMLAC) for photography and assistance with illustrations, M. Walsh (NHMLAC) for specimen preparation and B. Campbell (NHMLAC) for moulding and casting the material, all done at Marília. J. Velez-Juarbe (NHMLAC) assisted with the phylogenetic analyses, M. Fabbri, E. M. Steell and J. Benito provided feedback on methods and anatomy, P. H. Fonseca assisted with the computed tomography scans and I. Evangelista Jr (Comtur Marília) supported other logistical activities of the MPM. We thank A. Watanabe for assistance with the geometric morphometric analyses of the endocranium. G.N. and D.J.F. were funded by UKRI grant MR/X015130/1 to D.J.F. Fundação Carlos Chagas Filho de Amparo à Pesquisa do Estado do Rio de Janeiro (Proc. E-26/200.828/2021 to I.S.C.) and the Conselho Nacional de Desenvolvimento Científico e Tecnológico (CNPq 303596/2016 to I.S.C., Brazil) also provided support. More financial support was provided by various funds from the Dinosaur Institute (NHMLAC), including the Gretchen Augustyn Expedition Fund. A.G.M. is supported by CONICET PIBAA 1137 and ANID-Milenio-NCN2023_025. For the purpose of open access, the authors have applied a Creative Commons Attribution (CC-BY) licence to any author accepted manuscript version arising.

**Author contributions** L.M.C and G.N. are co-first authors on this publication. L.M.C. and A.G.M. established the original international collaboration. L.M.C., G.N. and D.J.F. conceived and designed the project. R.M.S. and I.S.C. assisted with domestic logistics of the international collaboration. L.M.C. and A.G.M. assisted with the curation of the fossil material. G.N., A.G.M., I.S.C. and Y.-H.W. assisted with computed tomography scanning. G.N. produced three-dimensional volumes of the individual bones and the endocranial and exocranial reconstructions. G.N. designed the analytical framework and conducted the geometric morphometric analyses. L.M.C. scored the morphological matrices and conducted the phylogenetic analysis. G.N. designed the figures with input from L.M.C. and D.J.F. L.M.C., G.N. and D.J.F. wrote the original manuscript with input from all the authors. L.M.C., I.S.C. and D.J.F. secured funding for the project.

**Competing interests** The authors declare no competing interests.

**Additional information**
**Correspondence and requests for materials** should be addressed to Luis M. Chiappe, Guillermo Navalón or Daniel J. Field.

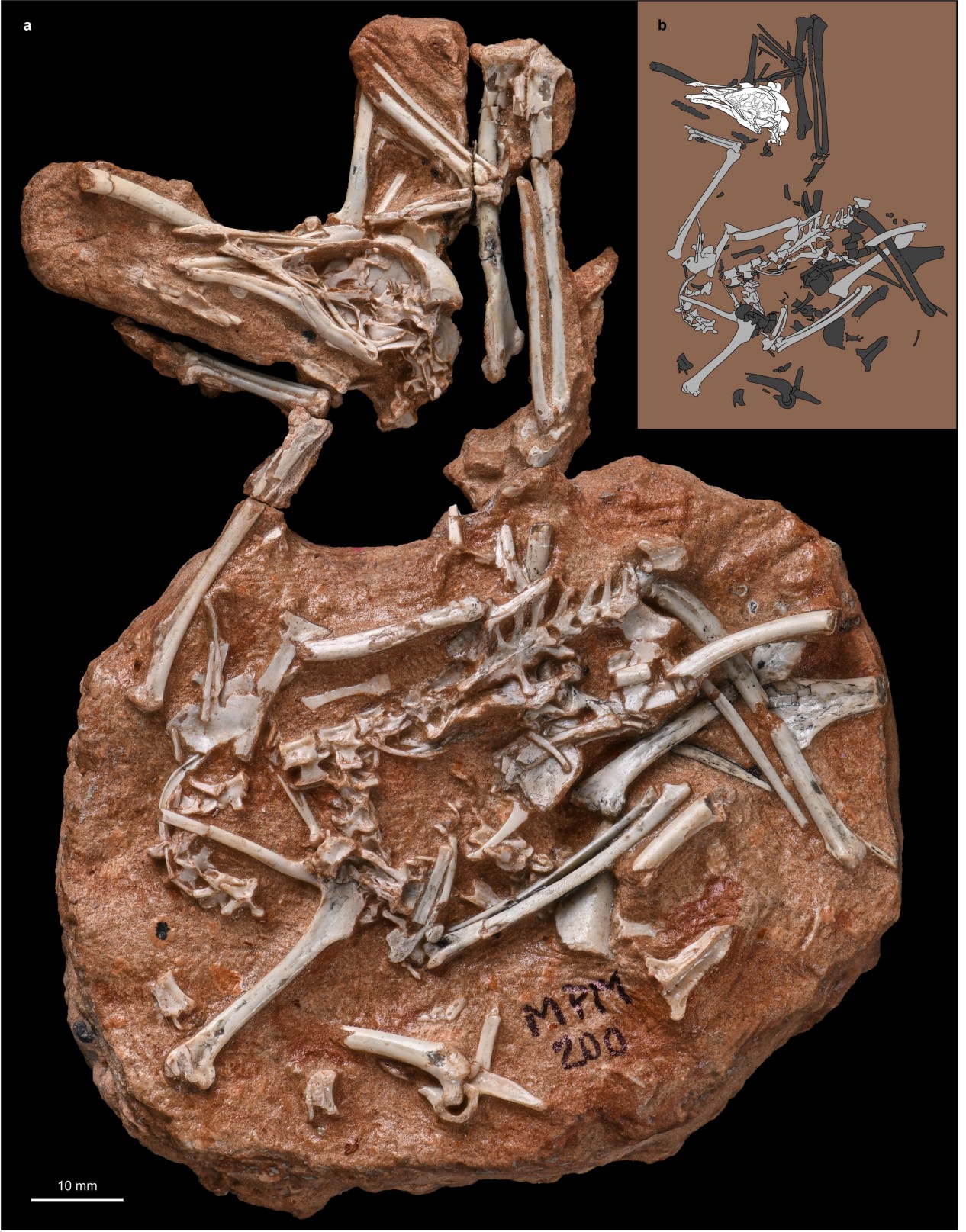

**Extended Data Fig. 1 | The holotype of the enantiornithine *Navaornis hestiae* gen. et sp. nov. and associated postcranial elements. a,b**, Photograph (**a**) and interpretative drawing (**b**) of the holotype (cranium, white colour in **b**, MPM-200-1), referred postcranial elements hypothesized to belong to the same individual (light grey in **b**, MPM-200), and additional associated bones interpreted as belonging to other individual enantiornithine birds (dark grey in **b**, MPM-200). The fossils were mechanically prepared, but their original connection and association is illustrated as they were found in the quarry.

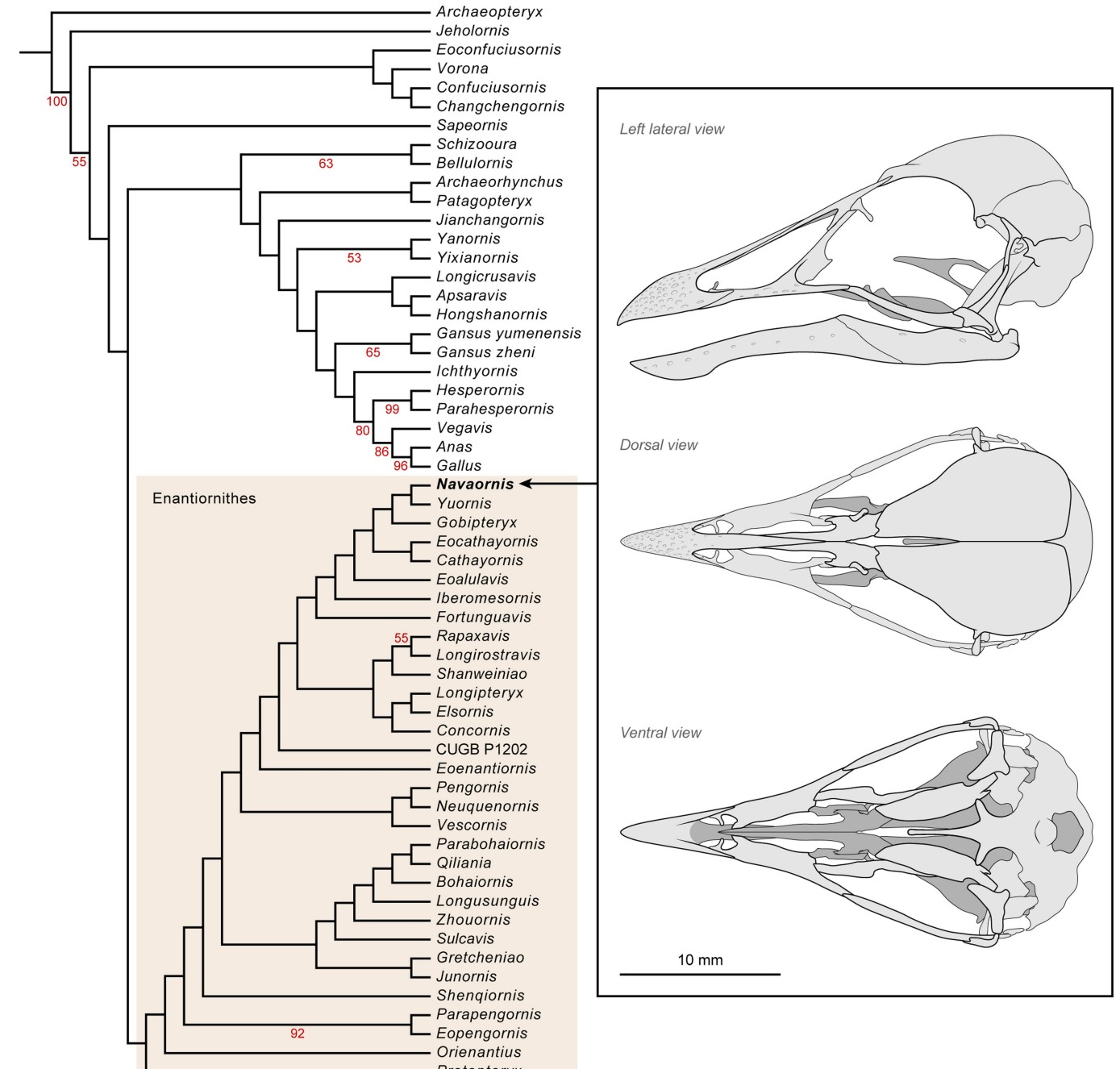

**Extended Data Fig. 2 | The phylogenetic position of *Navaornis hestiae*.**
Consensus tree based on implied weights (K = 12) using only the scorings
derived from the holotype (cranium, MPM-200-1). *Navaornis hestiae*
clusters with *Gobipteryx minuta* and *Yuornis junchangi* within Enantiornithes.

Bootstrap values over 50% are listed on branches. Inset shows 2D line drawings
based on our three-dimensional reconstruction of *N. hestiae* in dorsal, ventral
and left lateral views.

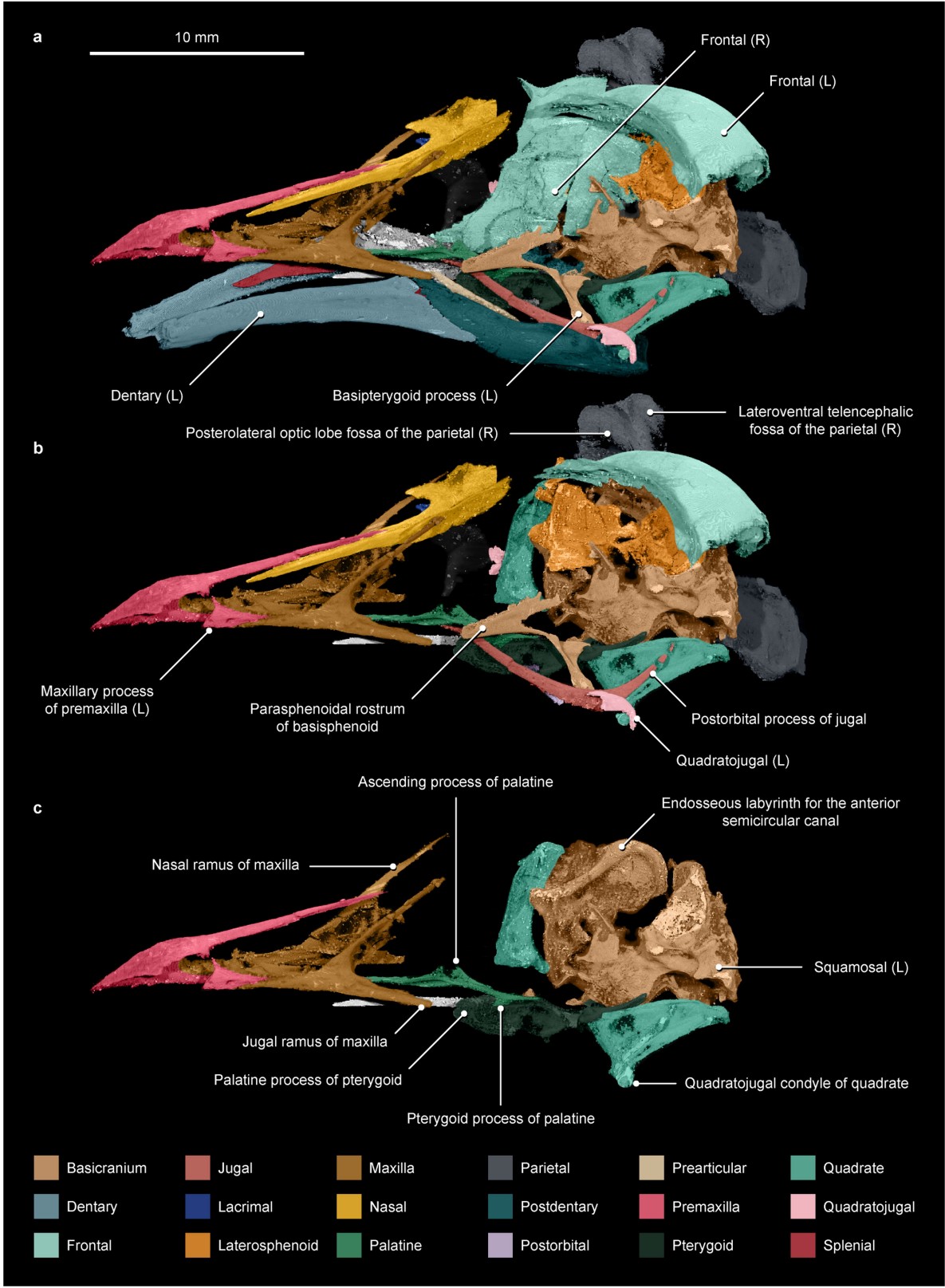

**Extended Data Fig. 3 | In-situ cranial anatomy of *Navaornis hestiae* (exposed side). a**, Digital render from the reconstructed three-dimensional volumes of the in-situ micro-CT scanned cranial elements of the holotype (MPM-200-1). **b,c**, Three-dimensional renders of the same region and view are shown with different cranial elements sequentially removed in order to expose key anatomical details of underlying bones. Cranial elements are colour-coded as in Fig. 1.

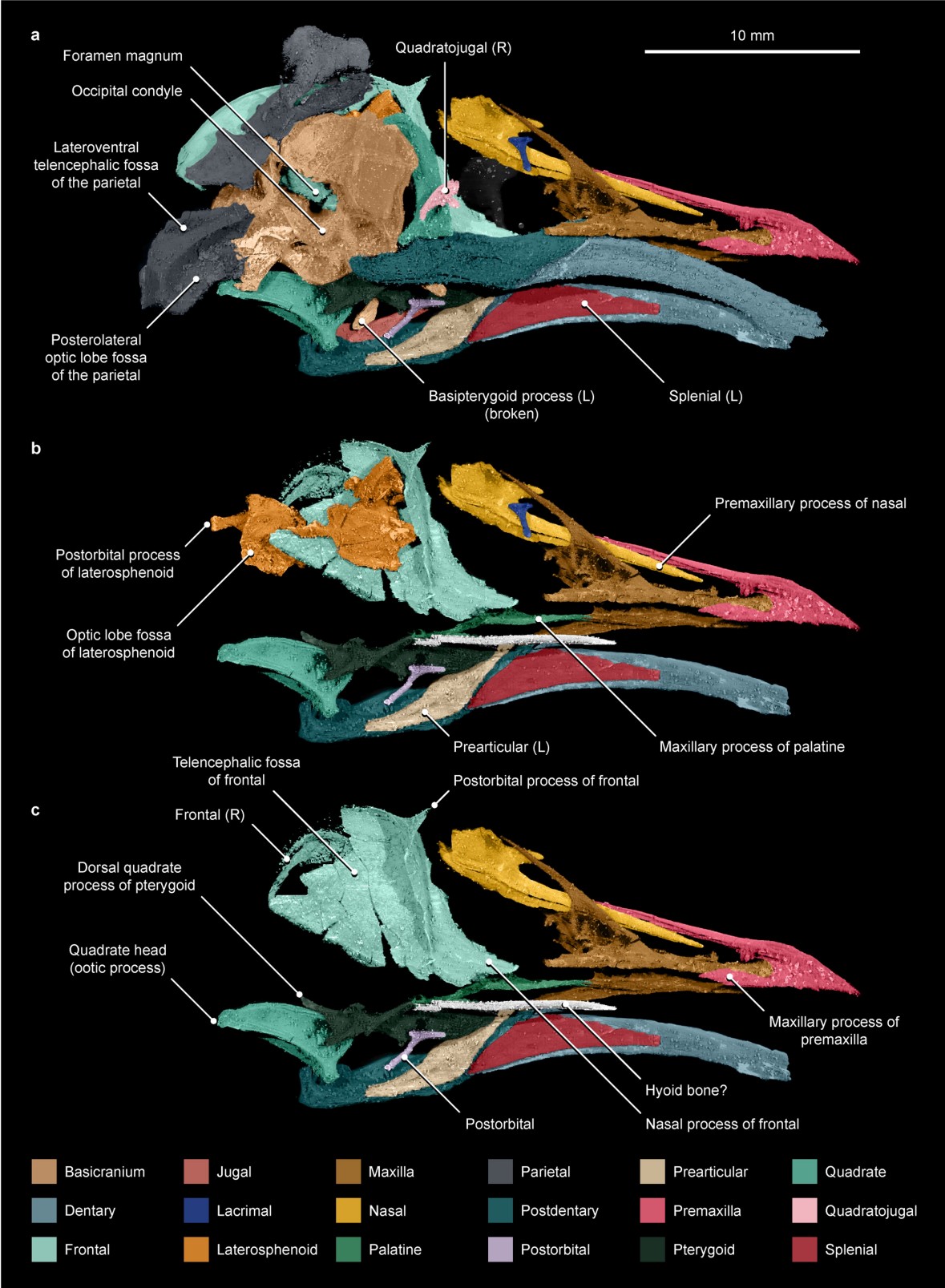

**Extended Data Fig. 4 | In-situ cranial anatomy of *Navaornis hestiae* (matrix side). a**, Digital render from the reconstructed three-dimensional volumes of the in-situ micro-CT scanned cranial elements of the holotype specimen (MPM-200-1). **b,c**, Three-dimensional renders of the same region and view are shown with different cranial elements sequentially removed in order to expose key anatomical details of underlying bones. Cranial elements are colour-coded as in Fig. 1.

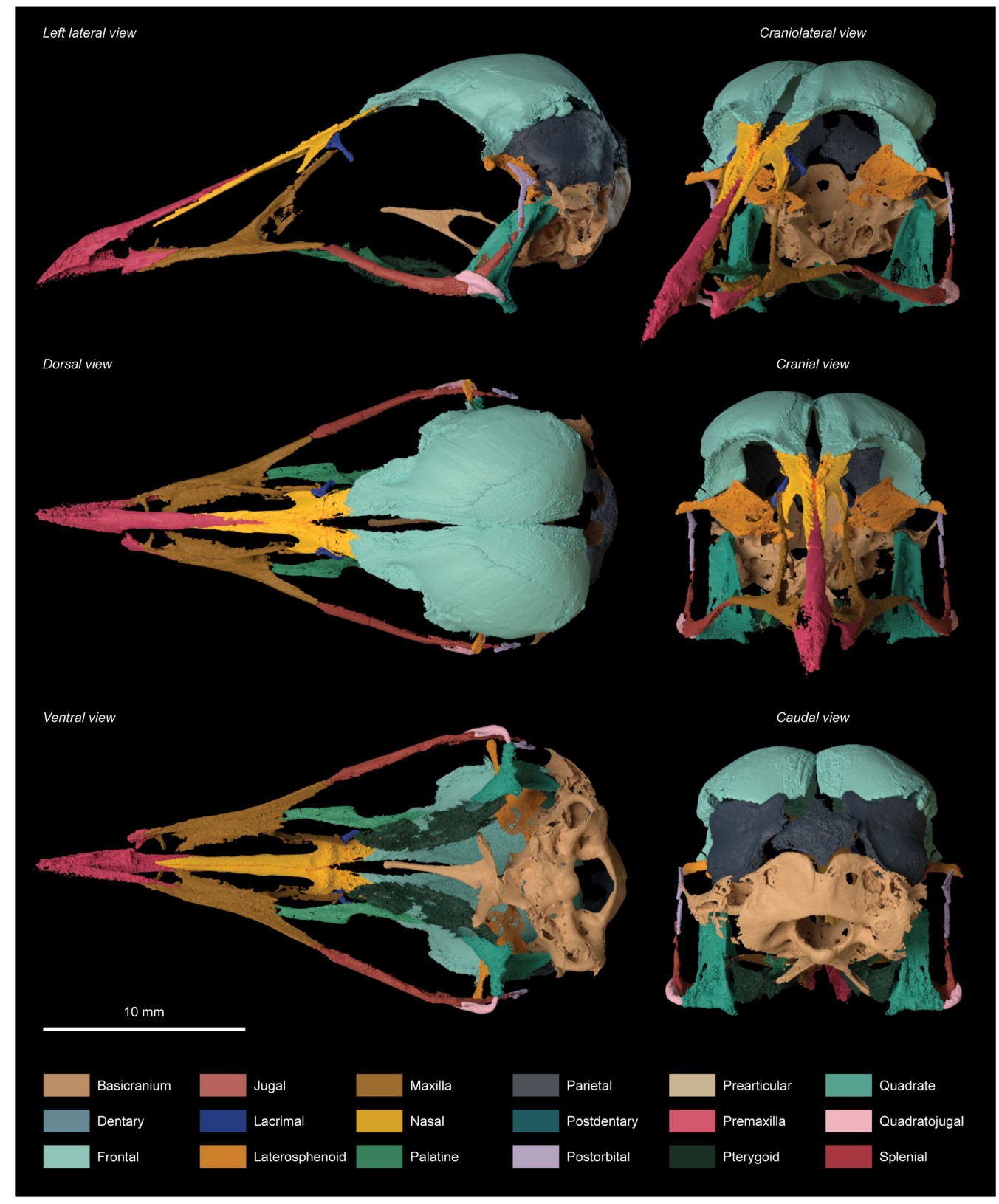

Left lateral view

Craniolateral view

Dorsal view

Cranial view

Ventral view

Caudal view

10 mm

| | Basicranium | | Jugal | | Maxilla | | Parietal | | Prearticular | | Quadrate |
| | Dentary | | Lacrimal | | Nasal | | Postdentary | | Premaxilla | | Quadratojugal |
| | Frontal | | Laterosphenoid | | Palatine | | Postorbital | | Pterygoid | | Splenial |

**Extended Data Fig. 5 | The three-dimensional cranial reconstruction of *Navaornis hestiae* (all main views).** Composite three-dimensional reconstruction of the skull of *Navaornis* from (MPM-200-1) and referred braincase (MPM-334-1) in six different views. Cranial elements are colour-coded as in Fig. 1.

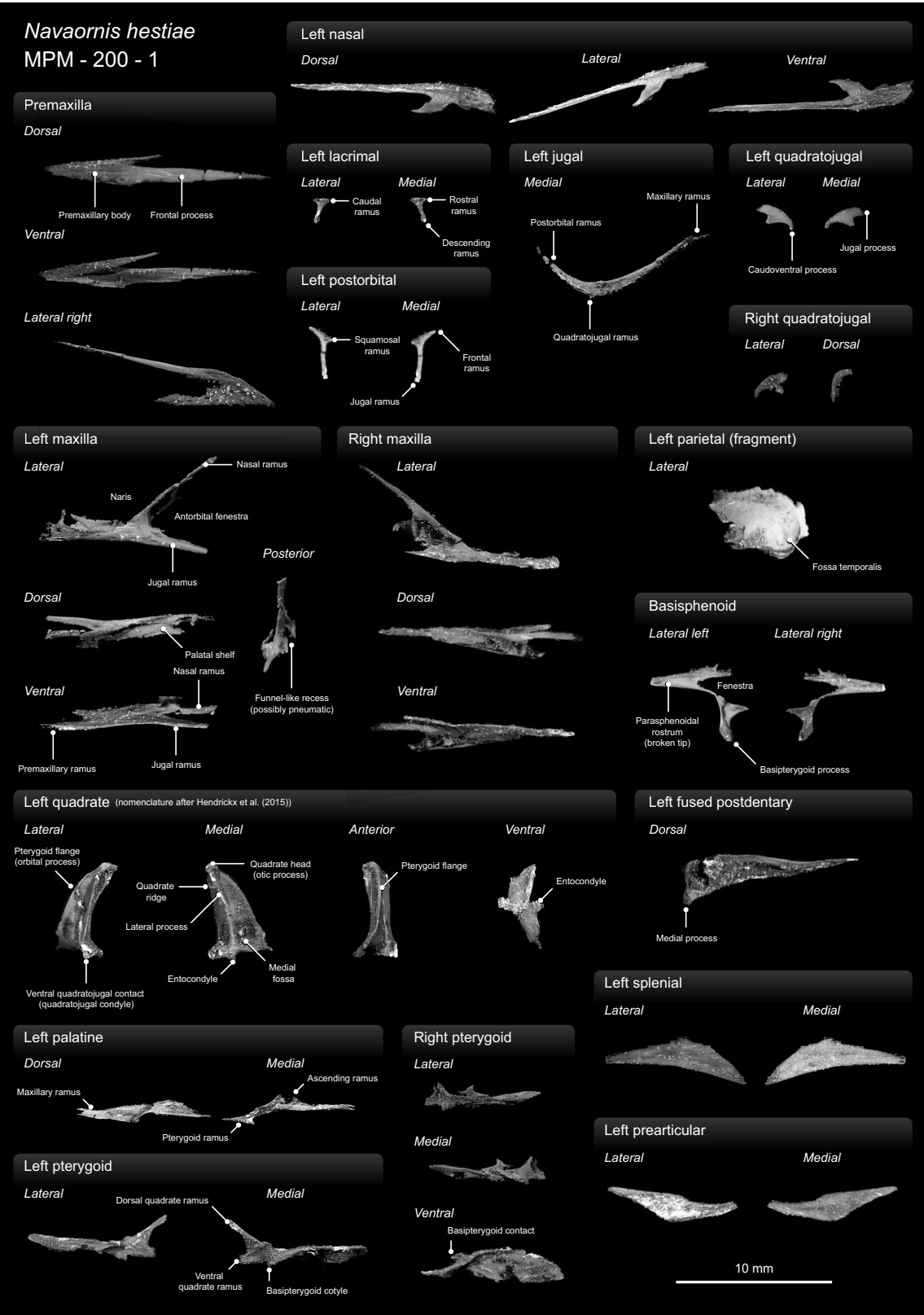

**Extended Data Fig. 6 | Detailed cranial anatomy of *Navaornis hestiae*.** Plate showing all the main cranial elements from the holotype specimen of *N. hestiae*. (MPM-200-1) in different views and annotated anatomical details commented on in the main text.

**a**  Three dimensional reconstruction of the braincase bones and their endocranial surfaces

*Dorsal*  *Caudal*

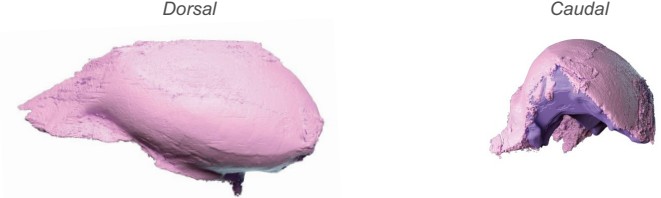

**b**  Retrodeformation of frontal and its endocast to match the articulating undistorted parietal

*Dorsal*  *Caudal*

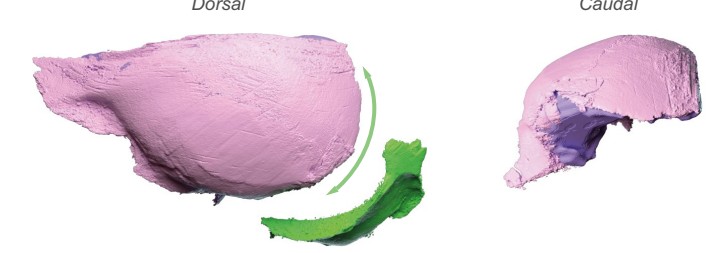

**c**  Articulation of all the braincase bones and their corresponding endocast surfaces

*Dorsal*  *Left lateral*  *Caudal*

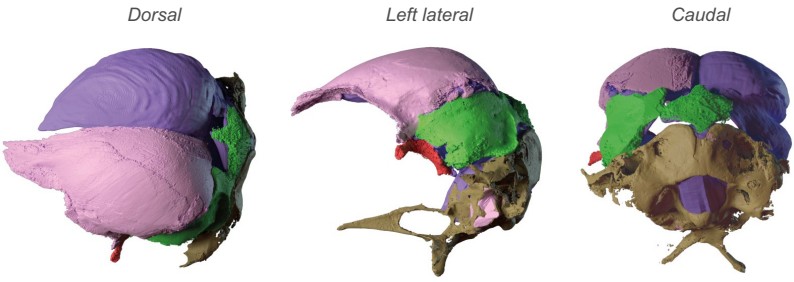

**d**  Removal of bone volumes leaving only the endocranial surfaces

*Dorsal*  *Left lateral*  *Caudal*

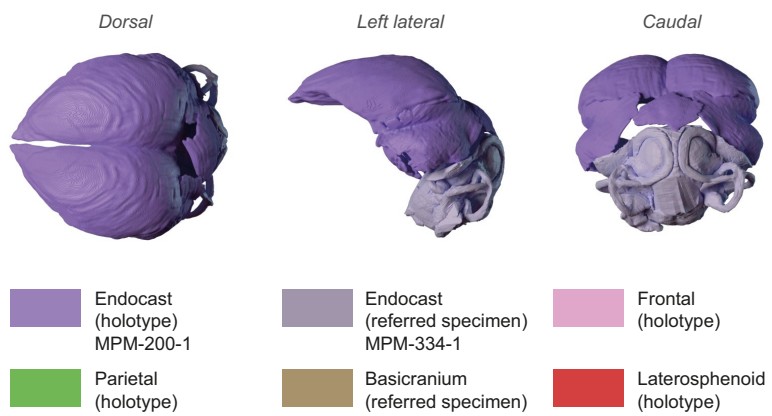

|  |  |  |
|---|---|---|
| Endocast (holotype) MPM-200-1 | Endocast (referred specimen) MPM-334-1 | Frontal (holotype) |
| Parietal (holotype) | Basicranium (referred specimen) | Laterosphenoid (holotype) |

**Extended Data Fig. 7 | The 3D reconstruction process of the brain and inner ear endocast of *Navaornis hestiae*.** Green arrow in **b** indicates the approximate outline of the dorsal rim of the parietal used to guide the retrodeformation process of the frontal.

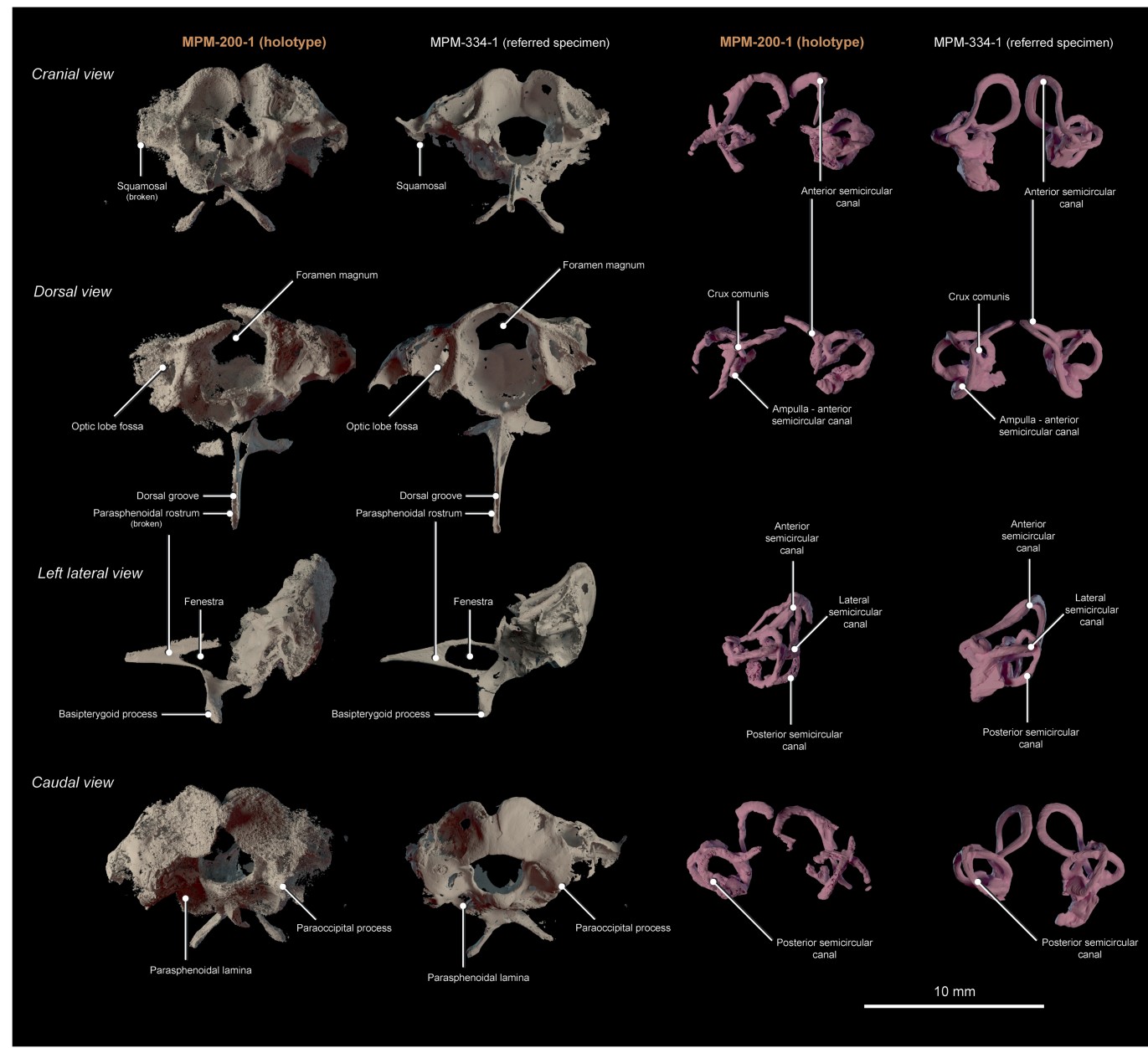

**Extended Data Fig. 8 | Isolated braincase (MPM-334-1) is morphologically indistinguishable from the holotype of *Navaornis hestiae*.** Comparison of the basicranium and the endosseous labyrinth of the inner ear between the holotype MPM-200-1 and the referred specimen MPM-334-1 of *Navaornis hestiae*.

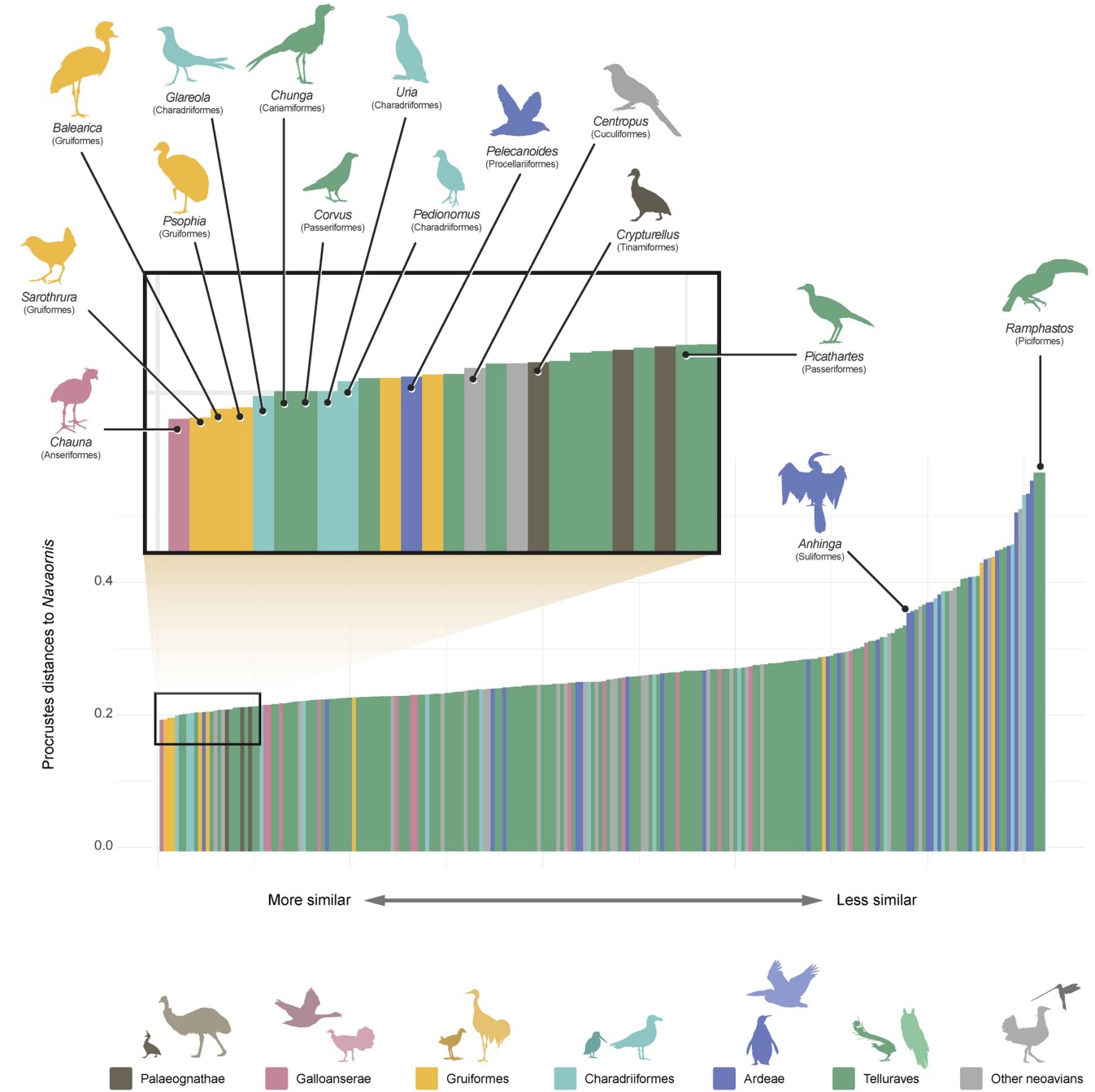

**Extended Data Fig. 9 | Cranial shape similarity between individual crown-bird species and *Navaornis hestiae*.** Shape differences are quantified as Procrustes distances between landmark configurations after Generalized Procrustes Superimposition (Procrustes residuals, see Methods). Species-values in the histogram plot are ordered from the most similar, to the least similar. Inset depicts the individual taxa that are closest in cranial shape to *Navaornis*. Note that the values of Procrustes distances to *Navaornis* are very similar among a wide range of species belonging to disparate lineages and that individual species do not necessarily resemble *Navaornis* in the same aspects of cranial geometry.

**a** Exocranium landmarking scheme
(Navalón et al. 2022)

**b** Endocranium landmarking scheme
(Watanabe et al., 2021)

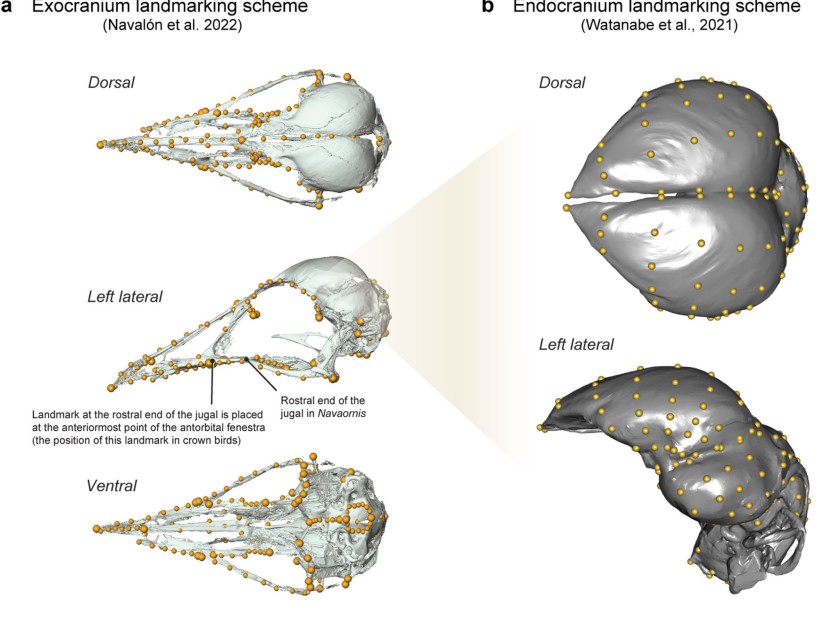

*Dorsal*

*Left lateral*

Landmark at the rostral end of the jugal is placed
at the anteriormost point of the antorbital fenestra
(the position of this landmark in crown birds)

Rostral end of the
jugal in *Navaornis*

*Ventral*

*Dorsal*

*Left lateral*

**c** PCA plots (exocranium)

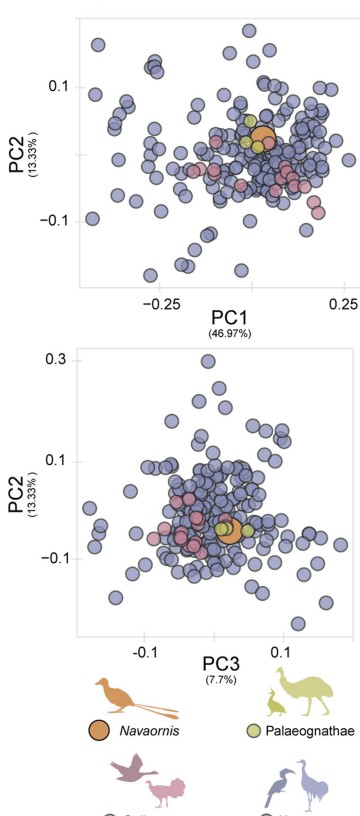

**d** PCA plots (endocranium)

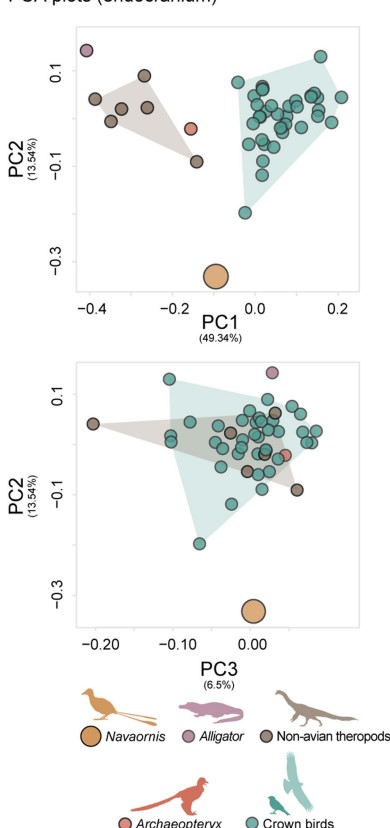

**Extended Data Fig. 10 | Additional information from the comparative geometric morphometrics analyses of exocranial and endocranial geometry. a**,**b**, Exocranial (**a**) and endocranial (**b**) landmarking scheme followed in this study. **a**, Some of the annotated changes in the exocranium landmarking scheme (see Methods for full list of changes). **c**,**d**, PCA bivariate plots of the morphospaces resulting from the three main axes (PC1-3) of exocranial (**c**) and endocranial (**d**) shape variation for *Navaornis* and the comparative datasets used in this study.

# Reporting Summary

## Statistics

For all statistical analyses, confirm that the following items are present in the figure legend, table legend, main text, or Methods section.

| n/a | Confirmed | |
|---|---|---|
| ☐ | ☒ | The exact sample size ($n$) for each experimental group/condition, given as a discrete number and unit of measurement |
| ☒ | ☐ | A statement on whether measurements were taken from distinct samples or whether the same sample was measured repeatedly |
| ☒ | ☐ | The statistical test(s) used AND whether they are one- or two-sided<br>*Only common tests should be described solely by name; describe more complex techniques in the Methods section.* |
| ☐ | ☒ | A description of all covariates tested |
| ☐ | ☒ | A description of any assumptions or corrections, such as tests of normality and adjustment for multiple comparisons |
| ☒ | ☐ | A full description of the statistical parameters including central tendency (e.g. means) or other basic estimates (e.g. regression coefficient) AND variation (e.g. standard deviation) or associated estimates of uncertainty (e.g. confidence intervals) |
| ☒ | ☐ | For null hypothesis testing, the test statistic (e.g. $F$, $t$, $r$) with confidence intervals, effect sizes, degrees of freedom and $P$ value noted<br>*Give P values as exact values whenever suitable.* |
| ☒ | ☐ | For Bayesian analysis, information on the choice of priors and Markov chain Monte Carlo settings |
| ☒ | ☐ | For hierarchical and complex designs, identification of the appropriate level for tests and full reporting of outcomes |
| ☒ | ☐ | Estimates of effect sizes (e.g. Cohen's $d$, Pearson's $r$), indicating how they were calculated |

*Our web collection on statistics for biologists contains articles on many of the points above.*

## Software and code

Policy information about availability of computer code

| | |
|---|---|
| Data collection | MPM-200-1 (holotype specimen for Navaornis hestiae gen. nov. sp. nov.) was scanned using a Bruker SkyScan 1173 CT scanner at the Instituto de Petróleo e dos Recursos Naturais (Laboratório de Sedimentologia e Petrologia) of the Pontifícia Universidade Católica do Rio Grande do Sul (PUCRS), Porto Alegre, Rio Grande do Sul, Brazil. |
| Data analysis | MPM-200-1 was scanned using a Bruker SkyScan 1173 CT scanner at the Instituto de Petróleo e dos Recursos Naturais (Laboratório de Sedimentologia e Petrologia) of the Pontifícia Universidade Católica do Rio Grande do Sul (PUCRS), Porto Alegre, Rio Grande do Sul, Brazil. Scanning parameters were as follows: 10.71 μm voxel size, 60 kV, 133 μA, exposure time 850 ms, averaging two frames, no 360° rotation, using an aluminium filter of 1.0 mm. Slices were reconstructed using the software NRecon v.1.7.4.6.<br><br>Volumes were digitally reconstructed and segmented using VGSTUDIOMAX 3.4.0 (VolumeGraphics).<br><br>Three-dimensional meshes of each recognisable segmented cranial element and endocranial surface were exported from VGSTUDIOMAX 3.4.0 to Blender 3.3.0, where they were rearticulated to reconstruct the skull and endocranial anatomy of Navaornis hestiae.<br><br>Owing to the remarkable lack of distortion of most cranial elements, only the left frontal bone required slight retrodeformation following established best practices ion Blender 3.3.0.<br><br>Landmark-based geometric morphometrics were used to quantitatively compare the exocranial and endocranial anatomy of Navaornis with crown birds and a selection of non-avian dinosaurs and Mesozoic birds. Exocranial landmarks were digitised in Avizo Lite 2019.3 (Thermo Fisher Scientific), and endocranial landmarks were digitised in Landmark Editor following previously described procedures. Thereafter, the landmark datasets were imported into the R statistical environment v4.1.2, where all downstream analyses were conducted (complete R code |

in Zenodo following this link: https://doi.org/10.5281/zenodo.10696014).

Phylogenetic analyses were conducted to ascertain phylogenetic relationships of the new taxon. Heuristic parsimony analyses were applied to a previously published and expanded (adding Navaornis and Yuornis) dataset using TNT v.1.670 under equal and implied weights (K = 3, K = 9, K =12).

For manuscripts utilizing custom algorithms or software that are central to the research but not yet described in published literature, software must be made available to editors and reviewers. We strongly encourage code deposition in a community repository (e.g. GitHub). See the Nature Portfolio guidelines for submitting code & software for further information.

## Data

Policy information about availability of data

All manuscripts must include a data availability statement. This statement should provide the following information, where applicable:
- Accession codes, unique identifiers, or web links for publicly available datasets
- A description of any restrictions on data availability
- For clinical datasets or third party data, please ensure that the statement adheres to our policy

Scan data and surface meshes of all preserved elements of Navaornis are housed on MorphoSource (morphosource.org; https://www.morphosource.org/projects/000608371/temporary_link/2vM9BSS4HaYcq1R2eMcxbTpc?locale=en). Phylogenetic matrices and morphometric landmark coordinates are provided at Zenodo (https://doi.org/10.5281/zenodo.10696014).

## Research involving human participants, their data, or biological material

Policy information about studies with human participants or human data. See also policy information about sex, gender (identity/presentation), and sexual orientation and race, ethnicity and racism.

| Reporting on sex and gender | Not Applicable |
|---|---|
| Reporting on race, ethnicity, or other socially relevant groupings | Not Applicable |
| Population characteristics | Not Applicable |
| Recruitment | Not Applicable |
| Ethics oversight | Not Applicable |

Note that full information on the approval of the study protocol must also be provided in the manuscript.

# Field-specific reporting

Please select the one below that is the best fit for your research. If you are not sure, read the appropriate sections before making your selection.

☐ Life sciences   ☐ Behavioural & social sciences   ☒ Ecological, evolutionary & environmental sciences

For a reference copy of the document with all sections, see nature.com/documents/nr-reporting-summary-flat.pdf

# Ecological, evolutionary & environmental sciences study design

All studies must disclose on these points even when the disclosure is negative.

| Study description | We report on a new fossil bird species from the Late Cretaceous of Brazil, whose complete skull is exceptionally well-preserved in three dimensions. This enabled the complete description of both the external (skull) and, remarkably, internal (brain) cranial morphology of this remarkable new fossil, yielding long-sought insight into how and when the unique modern bird brain evolved. |
|---|---|
| | Phylogenetic analyses resolve this new species as a member of the Mesozoic clade Enantiornithes, which split from the lineage leading to modern birds more than 130 million years ago. |
| | Landmark-based geometric morphometrics were used to quantitatively compare the exocranial and endocranial anatomy of Navaornis with crown birds and a selection of non-avian dinosaurs and Mesozoic birds. |
| | Our analyses allowed us to show that the skull of the new species displays an overall geometry that closely resembles that of modern birds, underscoring an unprecedented degree of convergence between these two distant avian lineages whereby archaic features of the skull acted as building blocks to yield a skull of surprisingly modern shape. |
| | Importantly, we present the first detailed endocranial description of a stem bird crownward of Archaeopteryx, clarifying the pattern and timing by which the distinctive neuroanatomy of living birds was assembled. |

| | |
|---|---|
| Research sample | MPM-200-1 (Museu de Paleontologia de Marília, Marília, São Paulo State, Brazil). A complete skull (Fig.1) articulated with the anterior-most cervical vertebrae, extracted from a block (MPM-200; Extended Data Figure 1) from the Sítio Paleontológico de Presidente bonebed. A cast of MPM-200 has been accessioned at the Dinosaur Institute, Natural History Museum of Los Angeles County. |
| Sampling strategy | NA |
| Data collection | MPM-200-1 was scanned using a Bruker SkyScan 1173 CT scanner at the Instituto de Petróleo e dos Recursos Naturais (Laboratório de Sedimentologia e Petrologia) of the Pontifícia Universidade Católica do Rio Grande do Sul (PUCRS), Porto Alegre, Rio Grande do Sul, Brazil.<br><br>Three-dimensional meshes of each recognisable segmented cranial element and endocranial surface were exported from VGSTUDIOMAX 3.4.0 to Blender 3.3.0, where they were rearticulated to reconstruct the skull and endocranial anatomy of Navaornis hestiae . Owing to the remarkable lack of distortion of most cranial elements, only the left frontal bone required slight retrodeformation following established best practices. This was accomplished using the 'Lattice' function in Blender, where mediolateral compression of this element was corrected to match the geometry of the dorsal rim of the complete and mostly undistorted left parietal bone. The same degree of retrodeformation was then applied to the endocast surface of the left frontal, that is, the left hemisphere of the telencephalon. The remaining surfaces composing the endocranium of Navaornis hestiae were: ventral (derived from the basicrania of both the holotype and referred specimen MPM-334-1), anterior (derived from the right and left laterosphenoids of the holotype), dorsal and lateral surfaces of the left optic lobes (derived from the parietal of the holotype), cerebellum (derived from the parietal of the holotype and basicranium of both the holotype and referred specimen MPM-334-1), and medulla (derived from both the holotype and referred specimen MPM-334-1). Left and right surfaces were mirrored from the best-preserved element/endocranial surface.<br><br>Landmark-based geometric morphometrics were used to quantitatively compare the exocranial and endocranial anatomy of Navaornis with a selection of pre-existing published data from crown birds and a selection of non-avian dinosaurs and Mesozoic birds.<br><br>Specifically, Generalised Procrustes Analysis were performed on both sets of landmark coordinates to separate shape data from size and other confounding factors, and the minimum bending energy criterion was used to slide curve (exocranium and endocranium) and patch (endocranium) semilandmarks following previously described procedures, using the function 'gpagen' in the R package geomorph v.4.0.5.<br><br>Principal Components Analyses were carried out on the exocranial and endocranial Procrustes coordinates to visualise shape variation using the function 'gm.prcomp' in geomorph.<br><br>To determine the extant species geometrically closest to Navaornis in exocranial shape, we determined the Procrustes distances between Navaornis and all extant taxa in our dataset (Extended Data Figure. 9) using Euclidean distances with the function 'dist' from the R package stats v.4.1.2.<br><br>Changes associated with major axes of exocranial and endocranial shape variation were illustrated as deformations warped from the three-dimensional surface of the exocranium and endocranium of the individual species closest to the mean shape in both samples. Specifically, this three-dimensional surface and the mean shape from the sample were projected onto the scores representing the 0.05 and 0.95 quantiles for each PC axis by means of thin-plate spline deformation using the function 'tps3d' from the package Morpho v.2.10 and 'shape.predictor' from geomorph. We also plotted the respective landmark configurations onto the deformed meshes using 'shape.predictor', and coloured these landmark constellations according to per-landmark-variances from each dataset using the 'hot.dots' function (freely available following this link: https://zenodo.org/record/3929193). |
| Timing and spatial scale | Not Applicable |
| Data exclusions | Not Applicable |
| Reproducibility | All the raw data and the methods are reported to ensure complete reproducibility. |
| Randomization | Not Applicable |
| Blinding | Not Applicable |

Did the study involve field work?  ☐ Yes  ☒ No

# Reporting for specific materials, systems and methods

We require information from authors about some types of materials, experimental systems and methods used in many studies. Here, indicate whether each material, system or method listed is relevant to your study. If you are not sure if a list item applies to your research, read the appropriate section before selecting a response.

## Materials & experimental systems

| n/a | Involved in the study |
|---|---|
| ☒ | ☐ Antibodies |
| ☒ | ☐ Eukaryotic cell lines |
| ☐ | ☒ Palaeontology and archaeology |
| ☒ | ☐ Animals and other organisms |
| ☒ | ☐ Clinical data |
| ☒ | ☐ Dual use research of concern |
| ☒ | ☐ Plants |

## Methods

| n/a | Involved in the study |
|---|---|
| ☒ | ☐ ChIP-seq |
| ☒ | ☐ Flow cytometry |
| ☒ | ☐ MRI-based neuroimaging |

# Palaeontology and Archaeology

| | |
|---|---|
| Specimen provenance | Referred specimens. MPM-334-1, an isolated basicranium from the Sítio Paleontológico de Presidente Prudente. MPM-200 includes a partially articulated postcranial skeleton, which is also referred to Navaornis hestiae and hypothesised to represent the same individual as MPM-200-1 (see Extended Data Figure 1). <br><br> Locality and Age. William's Quarry, Sítio Paleontológico, Presidente Prudente, São Paulo State, Brazil. The quarry is contained within the Adamantina Formation (Bauru Group, Bauru Basin); various lines of evidence suggest a late Santonian to early Campanian age (~85-75 million years ago) for this site. |
| Specimen deposition | MPM-200-1 (Museu de Paleontologia de Marília, Marília, São Paulo State, Brazil). |
| Dating methods | William's Quarry, Sítio Paleontológico, Presidente Prudente, São Paulo State, Brazil. The quarry is contained within the Adamantina Formation (Bauru Group, Bauru Basin); various lines of evidence14-17 suggest a late Santonian to early Campanian age (~85-75 million years ago) for this site. |

☒ Tick this box to confirm that the raw and calibrated dates are available in the paper or in Supplementary Information.

| | |
|---|---|
| Ethics oversight | No ethical approval or guidance from a specific institution was followed. |

Note that full information on the approval of the study protocol must also be provided in the manuscript.

# Plants

| | |
|---|---|
| Seed stocks | Not Applicable |
| Novel plant genotypes | Not Applicable |
| Authentication | Not Applicable |

