## [Peer Review File · Nature]

Manuscript Title: Cretaceous bird from Brazil informs the evolution of the avian skull and brain

Reviewer Comments & Author Rebuttals

Reviewer Reports on the Initial Version:

Referee #1 (Remarks to the Author):

The description of the skull and endocast of *Navaornis*, an enantiornithine, is incredibly important for those of us working on questions surrounding the origin of birds and provides a datapoint for a woefully underrepresented part of the avian stem lineage. I don't know that the results are unexpected based on the phylogenetic position of the fossil, but it is important nonetheless. The authors' assignment of the taxon to Enantiornithes appears to be well supported, thus it represents one of the only instances of three-dimensional preservation within this group (and pretty much from *Archaeopteryx* to *Ichthyornis*). The description of the specimen is well written and the reconstruction of the skull and endocast are impressive. I do think this is worth publication in *Nature* after the following concerns are addressed.

My main concern is with the discussion of the endocast. It is unclear what is meant by the expansion of the cerebellum within Enantiornithes (it would be nice to provide some sort of explanation). Is this interpretation based on the PCA shown in fig. 4B? Based on an admittedly qualitative assessment of this figure, PC1 looks to be describing primarily a mediolateral expansion of the cerebrum, which would have large scale effects on the rest of the brain. While there is undoubtedly a correlation between cerebrum expansion and cerebellar expansion, similar to what is seen in mammals, I don't know that I see evidence for a significant cerebellar expansion where they indicate on the tree. In fact, Watanabe et al. (2021), which used almost the exact same sampling of non-avian taxa, found that an "avian-type" cerebellum and optic tectum first appear further down the tree among non-avian theropods. Again in light of this, an explanation of what "cerebellar expansion" is referring to would be helpful. Figure 4B actually adds to this confusion because the endocast line drawings have the cerebellum marked as including the cerebellum, diencephalon, midbrain tegmentum, and the medulla. I realize that it is the color indicating "rest of brain" in 4A, but the cerebellum box in the character bars of 4B is the same color, which makes it seem like what is indicated in the line drawings is the cerebellum. The cerebellum should be distinguished from the rest of the brain to make this figure clearer. Also, make sure to include the flocculus as part of the cerebellum (unfortunately, the morphometric analysis of Watanabe does not include this part of the structure).

Following from the above comments, I also would argue that the "advanced" trait of ventrally displaced optic lobes is actually present within most maniraptorans. Once the cerebrum begins to expand somewhere around Maniraptora, the optic lobes begin to be displaced ventrolaterally, so this character has not historically been considered a crown group trait.

A few minor comments:

I realize this may be picky, but the use of the term “stem-ward” for taxa that are on the stem of the tree strikes me as wrong. I understand what the authors are saying, but maybe it would be better when talking about things on the stem to use terms that are relative to the crown group—i.e., crownward or further from the crown.

In the “Remarks” section, I don’t think that “(colloquially ‘birds’)” needs to be included since this is a term that is not defined and could potentially mean different things to different people.

On p. 9, I’m unclear what the authors mean by the “medial fusion of the telencephalic hemispheres”. Although there are commissures linking the hemispheres in living birds, they are not “fused”.

On p. 9, the Wulst actually comprises the hyperpallium. It doesn’t overlie it.

On p. 10, is the “medial bulge” running along the cerebellum referring to the occipital sinus? This is a character that can be present or absent in living birds and non-avian taxa. I’m not sure that it has anything to do with the degree of development of the cerebellum because it is also the sinus that forms the dural peak present in a lot of more basally diverging taxa such as tyrannosaurids. I don’t think it tells you much about its volumetric expansion.

From p. 11: Although smaller birds can exhibit fewer, shallower cerebellar folia, a recent paper (Cunha et al., 2021) found that the degree of foliation doesn’t affect the number of neurons within the cerebellum of crown birds.

Referee #1 (Remarks on code availability):

I am not qualified to review the code.

Referee #2 (Remarks to the Author):

It is great to see the report of *Navaornis*, a new Mesozoic enantiornithine bird which has important implications to our understanding of the evolution of birds. Complete Mesozoic bird fossils are very rare due to the delicate nature of birds, especially the 3D preserved skulls like what is reported in this manuscript. Enantiornithes is an important Mesozoic avian lineage, which went completely extinct during the Cretaceous–Paleogene mass extinction. The archaic skull of this new enantiornithine *Navaornis* combined with a crown bird-like toothless rostrum and brain flexion fills the gap between the stem-most early birds and the crown birds, providing values to clarify the long-lasting confusions about the evolution of avian skull.

This is an original and novel research, providing valid data and methodology and utilizing appropriate statistics. The conclusion is robust and valid. References credit appropriate to previous work. Summary, introduction and conclusions are clarified. However, here are several points that might be worthy to pay attention while the authors are revising this manuscript:

1. The geometric morphometric analysis of the skull only includes *Navaornis* and crown birds, so it will be plotted together with some crown birds anyway. The ideal way of testing if the skull of *Navaornis* is more similar to the crown birds compared to other Mesozoic birds, might be including more data of other Mesozoic birds and non-avian dinosaurs into this dataset, so that we could see the relative positions of *Navaornis*, crown birds, other Mesozoic birds and non-avian dinosaurs. However, the 3D data of the skull of the Mesozoic birds and non-avian dinosaurs are quite lacking now, so maybe a 2D geometric morphometric analysis including more published Mesozoic taxa could supplement the current analysis.
2. Line 234: The geometric morphometric analysis doesn't do any grouping analysis for habits or trophic ecologies, so I doubt a bit if only use procrustes distance to discuss or even mention the ecologies of the targeted taxon is meaningful and proper or not.
3. The morphological description part is a bit too long and it seems is now the only Results part. I would suggest shortening the description part a bit, by moving some details into the supplementary materials, such as the detailed descriptions of structures not that relevant to the discussion of avian skull evolution in this paper. In addition, the results of the quantitative analyses including the skull and the endocast could be expanded and moved to be another part of the Results, so that it will be clearer that this research includes both original qualitative and quantitative studies focusing on this new fossil bird.
4. The referred specimen MPM-334-1 is important to reconstruct the brain of *Navaornis*, but it is completely isolated from the holotype. The affiliation of this referred specimen may influence the analysis and discussions about the endocast of *Navaornis*, so the authors may want to add more explanations about it in the main text.

Here are also some minor suggestions:

1. Line 111: A ratio value could be provided here to quantitatively indicate that the maxilla constitutes a prominent component of the facial margin, since this has been emphasized more than one time in the manuscript.
2. Line 118: Is the medial shelf of the maxilla completely preserved in this specimen? I couldn't determine it confidently from the snapshot of the 3D models, so some texts could be added here to clarify it. The developing level of the medial shelf of the maxilla could be interesting to other researchers.
3. Line 121: The caudal margin of the naris is also formed by the maxilla and nasal in many other Mesozoic birds, such as Jeholornis and Sapeornis, so not that typical of only Enantiornithes, therefore maybe this sentence could be deleted.
4. Line 126: The lacrimal is extremely reduced compared to other enantiornithines. The authors may want to add some texts to confirm if it is preserved complete, and why it is identified as the lacrimal.
5. Line 139: The caudoventral contact between the quadrate and the quadratojugal is unusual, is it preserved in situ?
6. The 3D cranial reconstruction figure (Figure 2) is very pretty and informative. However, I think a fully retrodeformed 2D reconstruction with all the cranial elements could be added into this figure, especially for the lateral and palatal views. This may make it easier for the future researchers to cite and reuse the reconstruction figure while comparing Navaornis to other taxa.
7. Line 264: The similarity between the neuroanatomy of Navaornis and the extant bird embryos and hatchlings sounds very interesting, is it possible to add a figure in the supplementary to show this?

Author Rebuttals to Initial Comments:

We are happy to see that both Referees agree on the importance of our study and the exceptional nature of the new fossil specimen. We appreciate their feedback, which has strengthened and clarified our manuscript. Below we address their comments and feedback.

Referee #1 (Remarks to the Author):

Referee –The description of the skull and endocast of *Navaornis*, an enantiornithine, is incredibly important for those of us working on questions surrounding the origin of birds and provides a datapoint for a woefully underrepresented part of the avian stem lineage. I don't know that the results are unexpected based on the phylogenetic position of the fossil, but it is important nonetheless. The authors' assignment of the taxon to Enantiornithes appears to be well supported, thus it represents one of the only instances of three-dimensional preservation within this group (and pretty much from *Archaeopteryx* to *Ichthyornis*). The description of the specimen is well written and the reconstruction of the skull and endocast are impressive. I do think this is worth publication in Nature after the following concerns are addressed.

My main concern is with the discussion of the endocast. It is unclear what is meant by the expansion of the cerebellum within Euornithes (it would be nice to provide some sort of explanation). Is this interpretation based on the PCA shown in fig. 4B? Based on an admittedly qualitative assessment of this figure, PC1 looks to be describing primarily a mediolateral expansion of the cerebrum, which would have large scale effects on the rest of the brain. While there is undoubtedly a correlation between cerebrum expansion and cerebellar expansion, similar to what is seen in mammals, I don't know that I see evidence for a significant cerebellar expansion where they indicate on the tree. In fact, Watanabe et al. (2021), which used almost the exact same sampling of non-avian taxa, found that an "avian-type" cerebellum and optic tectum first appear further down the tree among non-avian theropods. Again in light of this, an explanation of what "cerebellar expansion" is referring to would be helpful.

Response – We agree that we have been rather ambiguous in regard to what we meant as 'cerebellar expansion'. To clarify, when we talk about 'cerebellar expansion' likely arising in Euornithes we mean the origin of cerebellar inflation/bulging that characterises crown and near crown birds - which admittedly might be related to changes in other regions of the brain (e.g., expansion of the telencephalon, as commented on by the reviewer, and/or changes in the flexion of the brain).

First, while the relative volume of the cerebellum (with respect to the total encephalic volume) may be comparable among some non-avian paravians, stem birds and crown, this is not the case with cerebellar volume relative to body mass, which appears to be significantly greater in crown birds than in any stem birds (see Balanoff et al., 2013, Figure S1c).

We also suggest that the cerebellum acquires a characteristic morphology near the avian crown (or somewhere within Euornithes). This is actually supported by the article the reviewer mentions. Watanabe et al. (2021) shows that crown birds occupy areas of cerebellar shape morphospace that are seemingly unique to them (Figure 2d in Watanabe et al., 2021) and only the unnamed troodontid IGM 100/1126 shows clear overlap in cerebellar shape with some crown birds. Similarly, there is no overlap in the shape of the optic lobes (Figure 2c) suggesting differences in shape between crown birds and taxa stemward to, and including, *Archaeopteryx*. A similar pattern can be seen in shape deviations from a common allometric pattern for the shape of both structures (Figure 3 in Watanabe et al., 2021).

These observations altogether support our own observations that the morphology and total size of the cerebellum and optic lobe are different between crown birds and *Archaeopteryx* together with more stemward taxa.

We have made the appropriate changes in the text to clarify our argument. However, we wish to emphasise that, although this aspect of our study represents the hypothesis that we consider to be best supported, it is nonetheless one that cannot be unambiguously confirmed at present; as such, we have tried to tone down our language to reflect that this hypothesis may well be challenged by future discoveries.

Referee – Figure 4B actually adds to this confusion because the endocast line drawings have the cerebellum marked as including the cerebellum, diencephalon, midbrain tegmentum, and the medulla. I realize that it is the color indicating “rest of brain” in 4A, but the cerebellum box in the character bars of 4B is the same color, which makes it seem like what is indicated in the line drawings is the cerebellum. The cerebellum should be distinguished from the rest of the brain to make this figure clearer. Also, make sure to include the flocculus as part of the cerebellum (unfortunately, the morphometric analysis of Watanabe does not include this part of the structure).

Response – We have modified this figure following the reviewer’s suggestion. The cerebellum (including the flocculus) is coloured in a different hue to the remainder of the brain.

Referee – Following from the above comments, I also would argue that the “advanced” trait of ventrally displaced optic lobes is actually present within most maniraptorans. Once the cerebrum begins to expand somewhere around Maniraptora, the optic lobes begin to be displaced ventrolaterally, so this character has not historically been considered a crown group trait.

Response – While taxa like the unnamed ‘troodontid’ and *Archaeopteryx* have some degree of expansion of the optic lobes, which lie somewhat more ventral to the telencephalon than in other more stemward taxa, we consider that *Navaornis* and crownward taxa still show a different degree of both expansion and ventralization of the optic lobes (bigger but also differently shaped as established by Watanabe et al., 2021). We believe this is clear in Figure 3b.

Referee – A few minor comments:

I realize this may be picky, but the use of the term “stem-ward” for taxa that are on the stem of the tree strikes me as wrong. I understand what the authors are saying, but maybe it would be better when talking about things on the stem to use terms that are relative to the crown group—i.e., crownward or further from the crown.

Response – We have made adequate changes throughout the text, though we wish to note that the application of the term ‘stemward’ is frequently used as a phylogenetic indicator of distance from a relevant crown group. For instance, *Archaeopteryx* is comparatively stemward of *Confuciusornis*, just as *Confuciusornis* is comparatively crownward of *Archaeopteryx*.

Referee –In the “Remarks” section, I don’t think that “(colloquially ‘birds’)” needs to be included since this is a term that is not defined and could potentially mean different things to different people.

Response – We have deleted this clarification.

Referee – On p. 9, I’m unclear what the authors mean by the “medial fusion of the telencephalic hemispheres”. Although there are commissures linking the hemispheres in living birds, they are not “fused”.

Response – We have modified our phrasing.

Referee –On p. 9, the Wulst actually comprises the hyperpallium. It doesn't overlie it.

Response – We meant that the Wulst, as a feature on the brain/endocast surface, overlies the hyperpallium as a volumetric region of the brain. We have made some changes to clarify this point following the way this structure is described in Balanoff and Bever (2020); an important reference in the field.

Referee – On p. 10, is the “medial bulge” running along the cerebellum referring to the occipital sinus? This is a character that can be present or absent in living birds and non-avian taxa. I'm not sure that it has anything to do with the degree of development of the cerebellum because it is also the sinus that forms the dural peak present in a lot of more basally diverging taxa such as tyrannosaurids. I don't think it tells you much about its volumetric expansion.

Response – We agree with the reviewer that we have been ambiguous in our description of cerebellar anatomy. We have made changes to hopefully help clarify this point. See our comment above for more details.

Referee – From p. 11: Although smaller birds can exhibit fewer, shallower cerebellar folia, a recent paper (Cunha et al., 2021) found that the degree of foliation doesn't affect the number of neurons within the cerebellum of crown birds.

Response – We were not aware of that recent paper, thank you very much. We have removed that statement from the text.

Referee #1 (Remarks on code availability):

I am not qualified to review the code.

Referee #2 (Remarks to the Author):

Referee – It is great to see the report of *Navaornis*, a new Mesozoic enantiornithine bird which has important implications to our understanding of the evolution of birds. Complete Mesozoic bird fossils are very rare due to the delicate nature of birds, especially the 3D preserved skulls like what is reported in this manuscript. Enantiornithes is an important Mesozoic avian lineage, which went completely extinct during the Cretaceous–Paleogene mass extinction. The archaic skull of this new enantiornithine *Navaornis* combined with a crown bird-like toothless rostrum and brain flexion fills the gap between the stem-most early birds and the crown birds, providing values to clarify the long-lasting confusions about the evolution of avian skull.

This is an original and novel research, providing valid data and methodology and utilizing appropriate statistics. The conclusion is robust and valid. References credit appropriate to previous work. Summary, introduction and conclusions are clarified. However, here are several points that might be worthy to pay attention while the authors are revising this manuscript:

Referee – The geometric morphometric analysis of the skull only includes *Navaornis* and crown birds, so it will be plotted together with some crown birds anyway.

Response—While we understand the reviewer’s reasoning (perhaps informed by the behaviour of a phylogenetic analysis), in the context of a landmark-based geometric morphometric analysis, if the skull of *Navaornis* were to be different in shape from crown birds such difference would be reflected in the Procrustes coordinates, Procrustes distances, or the PCA plots derived from them. GMM analyses are based on total phenotypic distances and any shape differences are reflected in the results, regardless of the inclusiveness of the dataset – for instance, one could include a single whale skull with a living bird dataset, and the analysis would show the very dissimilar shape between the whale and the birds).

Referee – The ideal way of testing if the skull of *Navaornis* is more similar to the crown birds compared to other Mesozoic birds, might be including more data of other Mesozoic birds and non-avian dinosaurs into this dataset, so that we could see the relative positions of *Navaornis*, crown birds, other Mesozoic birds and non-avian dinosaurs. However, the 3D data of the skull of the Mesozoic birds and non-avian dinosaurs are quite lacking now, so maybe a 2D geometric morphometric analysis including more published Mesozoic taxa could supplement the current analysis.

Response – We opted not to develop such an analysis because our knowledge of cranial anatomy in most Mesozoic birds comes from taphonomically flattened specimens, whose cranial reconstructions are subjective. As discussed above, from an analytical point of view, it is not necessary to develop another test to show the similar (or different) shape between the skull geometry of *Navaornis* and those of living birds, and we opted to limit our data to undistorted fossil skulls such as the one in the new specimen.

Referee – Line 234: The geometric morphometric analysis doesn't do any grouping analysis for habits or trophic ecologies, so I doubt a bit if only use procrustes distance to discuss or even mention the ecologies of the targeted taxon is meaningful and proper or not.

Response – We agree that this phrase could lead to ambiguities; we have made some changes in the text.

Referee –The morphological description part is a bit too long and it seems is now the only Results part. I would suggest shortening the description part a bit, by moving some details into the supplementary materials, such as the detailed descriptions of structures not that relevant to the discussion of avian skull evolution in this paper. In addition, the results of the quantitative analyses including the skull and the endocast could be expanded and moved to be another part of the Results, so that it will be clearer that this research includes both original qualitative and quantitative studies focusing on this new fossil bird.

Response – We have made some changes to streamline our anatomical description. However, we believe it is essential to provide a morphological description commensurate to the fact that we are also describing a new genus and species.

Referee – The referred specimen MPM-334-1 is important to reconstruct the brain of *Navaornis*, but it is completely isolated from the holotype. The affiliation of this referred specimen may influence the analysis and discussions about the endocast of *Navaornis*, so the authors may want to add more explanations about it in the main text.

Response – We have emphasised that MPM-334-1 is morphologically (in terms of anatomy, shape, and size) indistinguishable from the *Navaornis* braincase. MPM-334-1 was described in great detail in Chiappe, Navalón et al. (2022), and as such we didn't think it was necessary to add to the morphological description of our paper details already published in this publication.

Referee - Here are also some minor suggestions:

1. Line 111: A ratio value could be provided here to quantitatively indicate that the maxilla constitutes a prominent component of the facial margin, since this has been emphasized more than one time in the manuscript.

Response – We added a ratio in our description.

Referee – Line 118: Is the medial shelf of the maxilla completely preserved in this specimen? I couldn't determine it confidently from the snapshot of the 3D models, so some texts could be added here to clarify it. The developing level of the medial shelf of the maxilla could be interesting to other researchers.

Response—This feature can be observed in Extended Data Figure 6, referenced in the text. The new 2D reconstructions also show this morphology.

Referee – Line 121: The caudal margin of the naris is also formed by the maxilla and nasal in many other Mesozoic birds, such as Jeholornis and Sapeornis, so not that typical of only Enantiornithes, therefore maybe this sentence could be deleted.

Response – We have revised this sentence accordingly.

Referee – Line 126: The lacrimal is extremely reduced compared to other enantiornithines. The authors may want to add some texts to confirm if it is preserved complete, and why it is identified as the lacrimal.

Response—Added.

Referee – Line 139: The caudoventral contact between the quadrate and the quadratojugal is unusual, is it preserved in situ?

Response – The original positions of all elements are illustrated in multiple figures. The contact between the quadrate and the quadratojugal is not preserved completely *in situ* but these bones are preserved very close to its hypothesised articulation point. The morphology of these elements is clear from the well-preserved nature of the specimen.

Referee – The 3D cranial reconstruction figure (Figure 2) is very pretty and informative. However, I think a fully retrodeformed 2D reconstruction with all the cranial elements could be added into this figure, especially for the lateral and palatal views. This may make it easier for the future researchers to cite and reuse the reconstruction figure while comparing Navaornis to other taxa.

Response– We have included 2D line drawings illustrating cranial reconstructions in ventral, dorsal and left lateral views. These were added next to the phylogenetic tree in Extended Data Figure 2 and properly reference this in the main text.

Referee – Line 264: The similarity between the neuroanatomy of Navaornis and the extant bird embryos and hatchlings sounds very interesting, is it possible to add a figure in the supplementary to show this?

Response – These similarities can be observed by comparing our article with the article we referenced. The current structure of the article does not allow us to include a Figure such as this one as part of the main text or Extended Data Figures. We would be happy to add a Supplementary figure if requested by the Editor.

Reviewer Reports on the First Revision:

Referee #1 (Remarks to the Author):

I think, overall, the authors have done a very nice job of addressing my comments. I still have some questions about the cerebellum. Figure 2d in Watanabe et al. (2021) actually has both IGM 100/1126 and Archaeopteryx within the range of crown-group birds, which pulls that shift down to Paraves. Zanabazar technically plots outside of the minimum polygon for crown-group birds but not by much. The other non-avian taxa (apart from Allosaurus) are all oviraptorosaurs, which have a pretty distinct cerebellar morphology. You might also consider that if you draw a minimum polygon around the non-avian taxa in Fig. 2b that some living birds would be included within it. I honestly don't think this is overly problematic for the paper as a whole, which lays out a very cool story. I just want the authors to be aware that as more non-avian taxa are added to this analysis that their story might change.

Referee #2 (Remarks to the Author):

The quality of this manuscript is developed after the revisions – the usage of terms is clearer and the results of analyses are better illustrated. Sorry that I didn't make it clear in previous comments about the geometric morphometric analysis of the skull. I was not doubting the similarity between the skulls of Navaornis and crown birds, but was doubting a bit about the meaning of comparing them without any other Mesozoic bird sample included – it is possible that it is just hard to distinguish any Mesozoic bird from modern birds based on skull geometry. However, I do agree with the authors that the distorted fossil skulls of other Mesozoic birds might lead us to subjective results, so probably not a best timing to develop such analysis at this moment. Therefore I would suggest to accept this revised version, and look forward to see the publication of this interesting bird Navaornis.

Referee #2 (Remarks on code availability):

It seems that I am not qualified to download the codes.

Author Rebuttals to First Revision:

Referees' comments:

Referee #1 (Remarks to the Author):

I think, overall, the authors have done a very nice job of addressing my comments. I still have some questions about the cerebellum. Figure 2d in Watanabe et al. (2021) actually has both IGM 100/1126 and *Archaeopteryx* within the range of crown-group birds, which pulls that shift down to Paraves. *Zanabazar* technically plots outside of the minimum polygon for crown-group birds but not by much. The other non-avian taxa (apart from *Allosaurus*) are all oviraptorosaurs, which have a pretty distinct cerebellar morphology. You might also consider that if you draw a minimum polygon around the non-avian taxa in Fig. 2b that some living birds would be included within it. I honestly don't think this is overly problematic for the paper as a whole, which lays out a very cool story. I just want the authors to be aware that as more non-avian taxa are added to this analysis that their story might change.

Response: We agree with the reviewer that the differences in cerebellar volume and shape between crown and stem birds might not be completely clearcut, particularly considering our very patchy knowledge of brain evolution in birds. However, we still believe that current evidence suggests a volumetric expansion of the cerebellum in the lineage leading to the crown that postdates the divergence of more stemward taxa (e.g., *Navaornis*)—clearly, this interpretation may change as more 3D skulls of stem taxa are discovered. First, the relative volume of the cerebellum with respect to body mass appears to be significantly greater in crown birds than in any stem birds (see below Figure S1c from Balanoff et al., 2013). The only clear exception to this pattern is *Conchoraptor*, an oviraptorosaurian theropod, which has a very different cerebellar shape to even other stem birds (as indicated in Figure 2d in Watanabe et al., 2021, see below too). Second, Figure 2d in Watanabe et al. (2021) concerns only the shape of the cerebellum (not the size, or volumetric expansion that we talk about in our paper) but still clearly distinguishes crown birds and stem birds, with the exception of the unnamed troodontid. *Archaeopteryx* would sit along the line of a polygon encompassing all crown birds in that plot (see below). Both plots show, however, that most of the volumetric or shape disparity of crown birds is occupied solely by crown birds, suggesting there is a cerebellar crown condition. However, we definitely agree with the reviewer that these plots offer limited (although the only available) insights about the evolution of the brain in birds (for instance, Figure 2d from Watanabe et al., 2021 only represents a projection of about 60% of cerebellar shape variance) and adding more taxa in the future, from both crown and stem, will change our inferences about the evolutionary history of these traits. In light of these considerations we have modified our phrasing in the manuscript.

Figure S1c from Balanoff et al., 2013

Figure 2d in Watanabe et al. (2021). Convex hulls overly for clarity, hollow circle represents *Archaeopteryx*.

Referee #2 (Remarks to the Author):

The quality of this manuscript is developed after the revisions – the usage of terms is clearer and the results of analyses are better illustrated. Sorry that I didn't make it clear in previous comments about the geometric morphometric analysis of the skull. I was not doubting the similarity between the skulls of Navaornis and crown birds, but was doubting a bit about the meaning of comparing them without any other Mesozoic bird sample included – it is possible that it is just hard to distinguish any Mesozoic bird from modern birds based on skull geometry. However, I do agree with the authors that the distorted fossil skulls of other Mesozoic birds might lead us to subjective results, so probably not a best timing to develop such analysis at this moment. Therefore I would suggest to accept this revised version, and look forward to see the publication of this interesting bird Navaornis.

Response: We appreciate the reviewer's insights and we agree with them.

Referee #2 (Remarks on code availability):

It seems that I am not qualified to download the codes.

Response: Apologies for this inconvenience, our link in Zenodo (<https://doi.org/10.5281/zenodo.10696014>) should now be working